# Projected changes in Rhine River flood seasonality under global warming

Erwin Rottler[1], Axel Bronstert[1], Gerd Bürger[1], and Oldrich Rakovec[2,3]

[1]Institute of Environmental Science and Geography, University of Potsdam, Karl-Liebknecht-Straße 24–25, 14476 Potsdam, Germany
[2]UFZ-Helmholtz Centre for Environmental Research, Permoserstraße 15, 04318 Leipzig, Germany
[3]Faculty of Environmental Sciences, Czech University of Life Sciences Prague, Kamýcká 129, Praha – Suchdol, 165 00, Czech Republic

**Correspondence:** Erwin Rottler (rottler@uni-potsdam.de)

**Abstract.** Climatic change alters the frequency and intensity of natural hazards. In order to assess potential future changes in flood seasonality in the Rhine River Basin, we analyse changes in streamflow, snowmelt, precipitation, and evapotranspiration at 1.5, 2.0 and 3.0 °C global warming levels. The mesoscale Hydrological Model (mHM) forced with an ensemble of climate projection scenarios (five general circulation models under three representative concentration pathways) is used to simulate the present and future climate conditions of both, pluvial and nival hydrological regimes.

Our results indicate that future changes in flood characteristics in the Rhine River Basin are controlled by increases in antecedent precipitation and diminishing snow packs. In the pluvial-type sub-basin of the Moselle River, an increasing flood potential due to increased antecedent precipitation encounters declining snowpacks during winter. The decrease in snowmelt seems to counterbalance increasing precipitation resulting in only small and transient changes in streamflow maxima. For the Rhine Basin at Basel, rising temperatures evoke changes from solid to liquid precipitation, which enhance the overall increase in precipitation sums, particularly in the cold season. At gauge Basel, the strongest increases in streamflow maxima show up during winter, when strong increases in liquid precipitation encounter almost unchanged snowmelt-driven runoff. The analysis of snowmelt events for gauge Basel suggests that at no point in time during the snowmelt season, a warming climate results in an increase in the risk of snowmelt-driven flooding. Snow packs are increasingly depleted with the course of the snowmelt season. We do not find indications of a transient merging of pluvial and nival floods due to climate warming.

## 1 Introduction

Current climatic changes entail changes in the frequency and intensity of natural hazards. Among other things, rising temperatures reinforce heat waves (Meehl and Tebaldi, 2004; Della-Marta et al., 2007; Fischer and Schär, 2010) and dry spells (Blenkinsop and Fowler, 2007; Samaniego et al., 2018b; Grillakis, 2019) and more intense precipitation increases the risk posed by floods and land slides (Dankers and Feyen, 2008; Rojas et al., 2012; Alfieri et al., 2015; Crozier, 2010; Huggel et al., 2012). Fundamental changes are expected in snow-dominated regions (Hock et al., 2019); alpine climatic changes go along with declining seasonal snow packs (Steger et al., 2013; Beniston et al., 2018; Hanzer et al., 2018), thawing permafrost (Ser-

reze et al., 2000; Schuur et al., 2015; Elberling et al., 2013; Beniston et al., 2018) and retreating glaciers (Zemp et al., 2006; Huss, 2011; Radić and Hock, 2014; Hanzer et al., 2018). Those cryospheric changes, in turn, impact water availability in and outside mountain areas (Barnett et al., 2005; Stewart, 2009; Junghans et al., 2011; Viviroli et al., 2011). The European Alps, for example, are the source region of numerous large rivers that form the basis of the economic and cultural development in various cities and communities (Beniston, 2012).

Recent studies suggest that rapid climatic changes have already altered flood characteristics in river systems across Europe. For example, Blöschl et al. (2019) indicate that during 1950–2010, increasing rainfall and soil moisture led to higher river flood discharges in northwestern Europe, while decreasing rainfall together with higher evapotranspiration rates decreased flood discharge in southern parts of the continent. Detected trends in flood magnitudes seem to align with trends in the spatial extent of the floods (Kemter et al., 2020). A further distinction of floods depending on return period and catchment area enables a detailed investigation of processes generating floods (Bertola et al., 2020). Most important mechanisms driving flooding in Europe are extreme precipitation, snowmelt and soil moisture excess (Berghuijs et al., 2019).

In large and diverse river basins, such as the the Rhine River Basin, all relevant mechanisms generating riverine floods can be detected. The southern part of the basin is influenced by snowmelt from the Alps and therefore commonly classified as nival (Belz et al., 2007; Speich et al., 2015). The runoff of a nival hydrological regime is primarily controlled by the accumulation and melt of a seasonal snow cover. Hence, runoff is low during winter and high during summer. The main tributaries of the Rhine River are rainfall-dominated. Runoff is high during winter and low during summer. Flooding in the rainfall-dominated tributaries usually occurs in winter and is driven by large-scale advective precipitation (Pfister et al., 2004; Bronstert et al., 2007).

Investigating changes in runoff seasonality and flood-generating mechanisms is important to assess challenges in future water resources management. Previous investigations conducted in Switzerland (e.g., Horton et al., 2006; Addor et al., 2014; Brunner et al., 2019), Austria (e.g., Kormann et al., 2015, 2016; Hanzer et al., 2018), Norway (e.g., Vormoor et al., 2015, 2016) or the United States (e.g. Brunner et al., 2020a, b) point at changes in snowmelt- and rainfall-generated runoff. For the Rhine River, studies have indicated that changes in both nival and pluvial flow alter hydrological regimes and their high/low flow characteristics (e.g., Middelkoop et al., 2001; Belz et al., 2007; Hurkmans et al., 2010; Huang et al., 2013; Alfieri et al., 2015; Stahl et al., 2016; Thober et al., 2018; Marx et al., 2018; Huang et al., 2018). Projections of discharge attained using hydrological models proved key in the attempt to assess the impact of climatic changes.

The aim of the present study is to investigate future changes in rainfall- and snowmelt-induced flooding in the Rhine River. We use the mesoscale Hydrologic Model (mHM; Samaniego et al., 2010; Kumar et al., 2013) forced with an ensemble of climate projection scenarios (five general circulation models under three representative concentration pathways) to assess projected changes in streamflow, snowmelt, rainfall and evapotranspiration characteristics under 1.5, 2.0, and 3.0 °C global warming. Special focus is on the hypothesis of a transient merging of nival and pluvial flow regimes by climate change, which suggests that in a warmer world, earlier snowmelt-induced floods originating from the Alps might superimpose with more intense rainfall-induced runoff from pluvial-type tributaries, creating a new flood type with potentially disastrous consequences (Fig. 1).

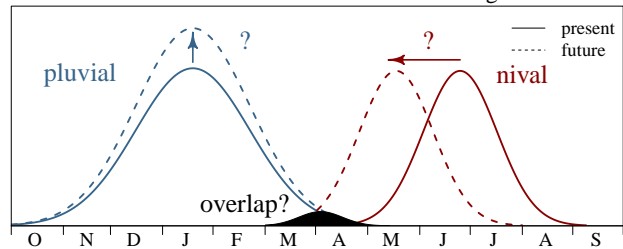

**Figure 1.** Idealised seasonal distribution of nival and pluvial flood frequencies and potential overlap due to climate change.

## 2 Data and Methods

### 2.1 Model set-up

The mesoscale hydrologic model (mHM) v.5.10 (Samaniego et al., 2010; Kumar et al., 2013; Samaniego et al., 2018a) is used to detect and assess projected changes in Rhine River floods under future climate conditions (Fig. 2 and 3). mHM is a spatially distributed hydrologic model based on grid cells. Key feature of mHM is the Multiscale Parameter Regionalization (MPR) technique, which allows to account for subgrid variability and provides simulations in seamless manner over multiple resolutions (e.g., Kumar et al., 2013; Rakovec et al., 2016; Samaniego et al., 2017). During MPR, high resolution physiographic land surface descriptors are translated into model parameters in the two phases of MPR, i.e., regionalization and upscaling. In the framework of this study, the high resolution physiographical datasets describing the main features of the terrain, e.g., digital elevation model, aspect, slope, soil texture, geological formation type, land cover and leave area index (LAI), are in 500 m resolution (Samaniego et al., 2019). The mHM model set-up distinguishes six soil layers up to a depth of 2 m based on Hengl et al. (2017). For each soil horizon the soil types are defined based on clay content, sand content and bulk density. We distinguish eight hydrogeological units. The baseflow recession parameters characterising each unit are determined during model calibration. Long-term climatologic monthly LAI maps are based on Mao and Yan (2019). Using a modified IGBP MODIS Noah classification scheme, 23 LAI classes are distinguished, whereby classes representing croplands, grassland, coniferous forest, mixed forest and mosaics of cropland and natural vegetation being the most common classes in the basin. More information on physiographical datasets, the mapping on a common 500 m × 500 m spatial resolution and underlying data sources is presented in Samaniego et al. (2019). All dominant hydrological processes are modelled at 5 km spatial resolution.

Meteorological forcing data of the model consists of daily average, maximum and minimum temperature and precipitation. Observational data sets are based on the E-OBS v12 gridded data sets (Haylock et al., 2008). Climate model data originates from the Inter-Sectoral Impact Model Intercomparison Project (ISI-MIP) (Hempel et al., 2013a, b; Warszawski et al., 2014). ISI-MIP bases on Global Climate Model (GCM) runs performed during the fifth phase of the Coupled Model Intercomparison Project (CMIP5; Taylor et al., 2012). Within ISI-MIP, daily data from five Global Climate Models (GCMs), i.e., GFDL-ESM2M, HadGEM2-ES, IPSL-CMSA-LR, MIROC-ESM-CHEM, NorESM1-M, were bias corrected and bi-linearly interpolated to a 0.5° × 0.5° grid. Bias correction of climate model data represents an indispensable step in climate change impact modelling

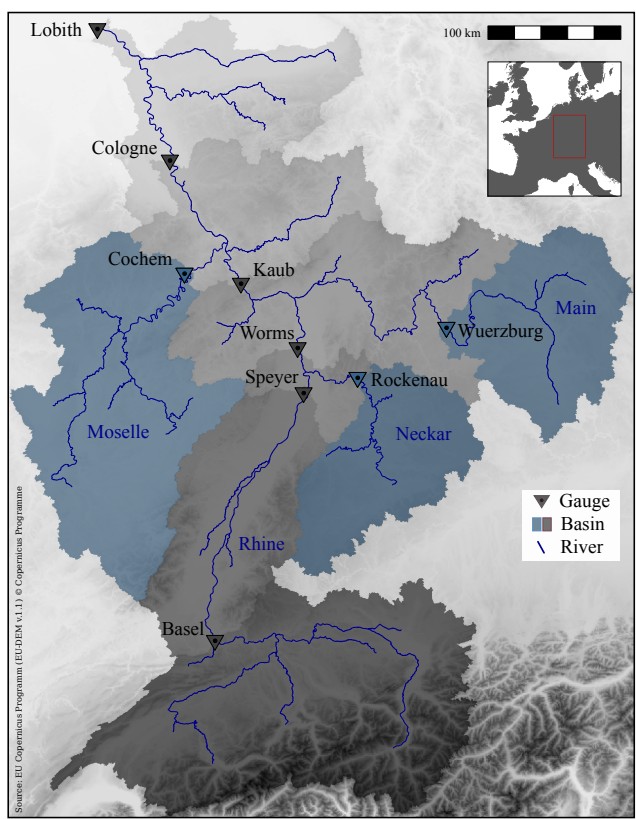

**Figure 2.** Topographic map of the Rhine River Basin at gauge Lobith with locations of all gauges and sub-basins investigated.

applications. Systematic deviation, e.g., due to imperfect model representations of atmospheric processes or errors in the parameterisation chain, need to be corrected (Ehret et al., 2012). A detailed description of the trend-preserving statistical bias correction method developed and applied within ISI-MIP, which includes an additive correction approach for temperature and a multiplicative correction for precipitation, is presented in Hempel et al. (2013b). GCM data used cover the period 1950–2099

5    and include three representative concentration pathways (RCPs) 2.6, 6.0 and 8.5. In the framework of the project "EDgE - End-to-end Demonstrator for improved decision making in the water sector in Europe" by order of the Copernicus Climate Service (edge.climate.copernicus.eu), meteorological data sets were interpolated to a 5 km grid using external drift kriging (Samaniego et al., 2019).

mHM forced with E-OBS meteorological data is calibrated for the Rhine Basin at gauge Lobith against observed streamflow

10   at the three gauges Lobith, Basel and Cochem during 1951–1975 using the Dynamically Dimensioned Search algorithm (DDS; Tolson and Shoemaker, 2007) and the Nash-Sutcliffe efficiency (NSE; Nash and Sutcliffe, 1970). In the framework of this multi-basin calibration, we simultaneously optimise NSE values for the three gauges and attain one set of global parameters, which we apply to the entire basin. We use a multi-basin approach to ensure that rainfall and snowmelt triggered runoff from both nival and pluvial dominated sub-basins as well as streamflow in the main channel of the Rhine River are considered during

**Table 1.** River gauges investigated: Location (WGS 84), GRDC identification number, catchment area, Nash-Sutcliffe efficiency (NSE) and Kling-Gupta efficiency (KGE) between observed and modelled runoff (NSE / KGE). The model has been calibrated against observation from the three gauges (Lobith, Basel and Cochem) with the NSE as objective function during 1951–1975.

| Name | GRDC-ID | Lat. | Lon. | Area (km$^2$) | 1951–1975 | 1976–2000 | 1951–2000 |
|------|---------|------|------|-----------|-----------|-----------|-----------|
| Lobith | 6435060 | 51.840 | 6.110 | $1.61 \cdot 10^5$ | 0.91 / 0.93 | 0.90 / 0.89 | 0.91 / 0.91 |
| Cologne | 6335060 | 50.937 | 6.963 | $1.44 \cdot 10^5$ | 0.92 / 0.96 | 0.92 / 0.94 | 0.92 / 0.95 |
| Cochem | 6336050 | 50.143 | 7.168 | $2.71 \cdot 10^4$ | 0.84 / 0.75 | 0.87 / 0.77 | 0.85 / 0.77 |
| Kaub | 6335100 | 50.085 | 7.765 | $1.03 \cdot 10^5$ | 0.90 / 0.90 | 0.92 / 0.92 | 0.91 / 0.91 |
| Wuerzburg | 6335500 | 49.796 | 9.926 | $1.40 \cdot 10^4$ | 0.73 / 0.81 | 0.79 / 0.84 | 0.76 / 0.83 |
| Worms | 6335180 | 49.641 | 8.376 | $6.89 \cdot 10^4$ | 0.85 / 0.87 | 0.88 / 0.90 | 0.87 / 0.88 |
| Rockenau | 6335600 | 49.438 | 9.005 | $1.27 \cdot 10^4$ | 0.75 / 0.74 | 0.74 / 0.71 | 0.74 / 0.73 |
| Speyer | 6335170 | 49.324 | 8.449 | $5.31 \cdot 10^4$ | 0.82 / 0.88 | 0.86 / 0.90 | 0.84 / 0.89 |
| Basel | 6935051 | 47.559 | 7.617 | $3.59 \cdot 10^4$ | 0.71 / 0.83 | 0.75 / 0.85 | 0.73 / 0.84 |

calibration. MPR enables the sampling in a lower-dimensional space, in turn, speeding up the convergence of the optimization algorithm (Samaniego et al., 2010). In total, we calibrate 47 global parameters using 1000 model iterations. A detailed overview of global parameters and their linkage with basin predictors in the regionalization transfer functions are presented in Samaniego et al. (2010) and Kumar et al. (2013). In order to evaluate the model performance in all important sub-regions of the entire Rhine River, the mHM performance is evaluated at additional six independent gauges (Fig. 2) and during an independent evaluation period (1976–2000) using the NSE and the Kling-Gupta-Efficiency (KGE; Gupta et al., 2009) (Table 1). Analyses evaluating streamflow simulations for the historic time frame 1951–2000 are given in the Appendix (Fig. A1, B1 and C1). Similar to investigations presented in the supplementary material of Thober et al. (2018), we assess streamflow maxima and the 90 % streamflow quantile of the hydrological year. In addition, we evaluate the timing of annual streamflow maxima and 90 % streamflow quantiles on a monthly basis. All observational discharge times series are obtained from the Global Runoff Data Centre (GRDC).

The multiscale Routing Model (mRM; Thober et al., 2019) is used for routing river runoff using the adaptive time step scheme (aTS). The kinematic wave equation (Lighthill and Whitham, 1955), a simplification of the Saint-Venant equation (de Saint-Venant, 1871), is solved using a finite difference scheme. The kinematic wave equation only needs little information on the river topography and assesses the advection and the attenuation of flood waves. The time step selected within aTS only depends on the spatial resolution and is independent of the temporal resolution of the meteorological forcing. In our model set-up, water is routed through the river network at a temporal resolution of 30 min. The high-resolution river network is based on a 500 x 500 m digital elevation map and is upscaled to operate on a 5 km routing resolution. Within the upscaling process, the flow direction in the lower resolution (routing resolution) is equal to the flow direction in the underlying high-resolution

grid cell with the highest flow accumulation (Samaniego et al., 2010). The stream celerity is determined as a function of terrain slope (Thober et al., 2019).

All dominant hydrological processes are modelled at 5 km spatial resolution. We estimate reference crop evapotranspiration following the Hagreaves-Samani equation, an empirical approach using minimum climatological data (Hargreaves and Samani, 1985; Samani, 2000). The empirical coefficient of the equation is determined during calibration. The usage of this simple approach enables a consistent set-up across historical and future model space. The actual evapotranspiration is estimated based on the fraction of roots in the soil horizons and a stress factor for reducing potential values calculated based on the actual soil moisture. The stress factor is determined using the Feddes equation (Feddes et al., 1976). If the soil moisture is below the permanent wilting point, evapotranspiration is reduced to zero. In case the soil moisture is above field capacity, the evapotranspiration equals the fraction of roots. If the soil moisture is in between the permanent wilting point and field capacity, evapotranspiration is reduced by the fraction of roots times the stress factor. The mHM set-up distinguishes six soil layers up to a total depth of 2 m. Organic matter is possible until 0.3 m. In total, more than 2000 soil types with different clay content, sand content and bulk density are defined. Land surface with impervious cover are treated as free-water surfaces and actual evapotranspiration is estimated with an additional evaporation coefficient. More details of the soil parameterization in mHM can be found in Livneh et al. (2015).

The canopy interception is modelled with a maximum interception approach. The maximum interception capacity is estimated based on the given LAI values. Water can leave the interception storage as throughfall, which is estimated as a function of the current and maximum canopy water content and the incoming precipitation. Evaporation from the canopy storage depends on the current and maximum canopy water content and the potential values of evapotranspiration. We simulated snow using an empirical degree-day approach, whereas degree-day-factors differ depending on the dominant land use class. In order to account for snowmelt following the energy input from liquid rainfall, degree-day factors are increased depending on the amount of liquid precipitation. Degree-day factors only can increase to a certain threshold value. Due to the spatial resolution of 5 km, our model set-up does not capture the highest elevations in the basin. To also capture the snow dynamics at mountain peaks, meteorological input data would need to be at higher spatial resolution and more advanced snow/ice processes would need to be considered. Surface runoff from impervious areas is calculated based on a linear reservoir exceedance approach. Interflow from the unsaturated zone is determined using a nonlinear reservoir with saturation excess. Groundwater is assumed as a linear reservoir. mHM does not included glacier and lake modules yet.

The changes in mHM-based flood seasonality are further differentiated and scrutinised for three different warming levels: 1.5, 2.0 and 3.0 °C. Within each future model run, the 30-year time windows when the warming levels (compared to the historic time window 1971–2000) are reached, are determined. The period 1971–2000 is assumed to be warmer by 0.46 °C compared to pre-industrial levels already (Vautard et al., 2014). For example, when comparing 30-year running temperature means from the IPSL-CM5A-LR model run under RCP 6.0, temperatures reach 1.5 °C warming compared to pre-industrial levels in the 30-year time window 2009–2038, 2.0 °C warming during 2028–2057 and a 3.0 °C warming in the period 2066–2095. 14 GCM-RCP realisations reach 1.5 °C, 13 reach 2.0 °C, and 8 reach 3.0 °C global warming. A detailed description of the determination

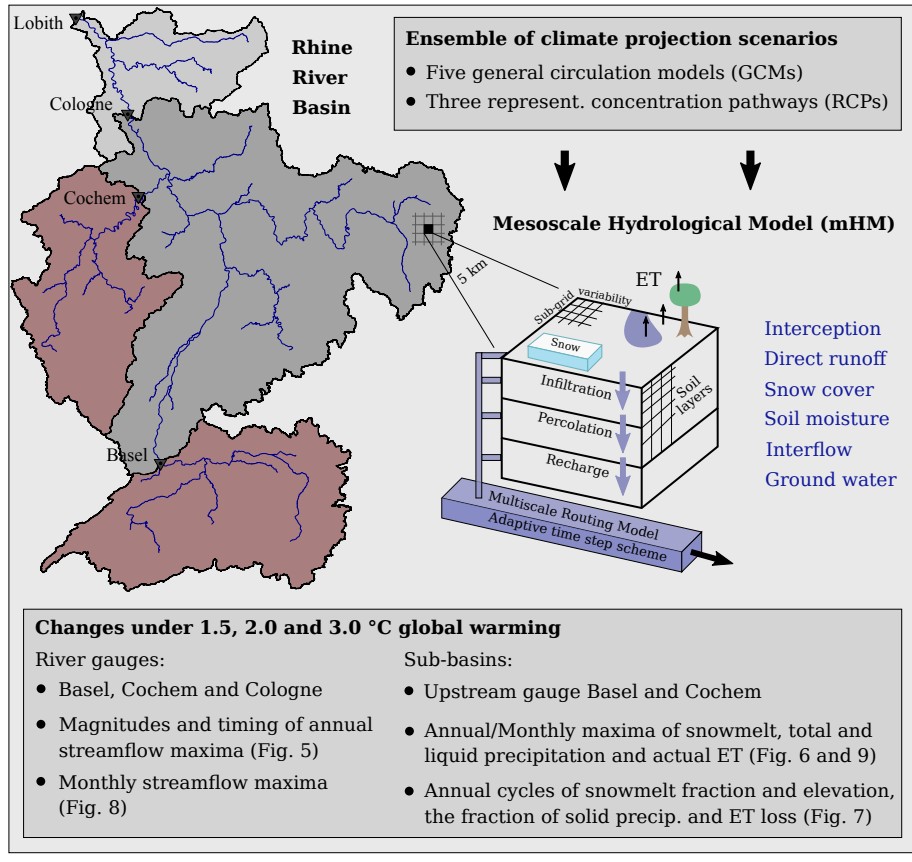

**Figure 3.** Scheme of the analytical set-up depicting gauges (Basel, Cochem and Cologne) and sub-basins (at gauges Basel and Cochem) investigated in detail.

of warming levels including a table with 1.5, 2.0 and 3.0 °C time periods of GCM-RCP realisation (Table S1) is given in the supplementary material of Thober et al. (2018).

## 2.2 Changes in streamflow characteristics

In order to assess the changes in flood characteristics, we determine the timing and magnitude of annual and monthly maxima of streamflow, precipitation (total and liquid), snowmelt and actual evapotranspiration for the hydrological year starting on the 1st of October (Tab. 2). In case of precipitation, we investigate maxima of 5-day sums ($P_{max5}$). Investigating thousands of annual streamflow maxima for different Swiss catchments with regard to flood-triggering precipitation, Froidevaux et al. (2015) conclude that precipitation 2 to 3 days before an event is an important determinant of flood magnitude. To account for larger catchment sizes and hence longer travel times in our study catchments, we chose a five 5-day window. For snowmelt and evapotranspiration, we extend this time window to 10 days and assess the magnitude and timing of 10-day sums ($S_{max10}$ and $ET_{max10}$). We assume that in order to have substantial impact on streamflow, meteorological conditions favouring snowmelt or

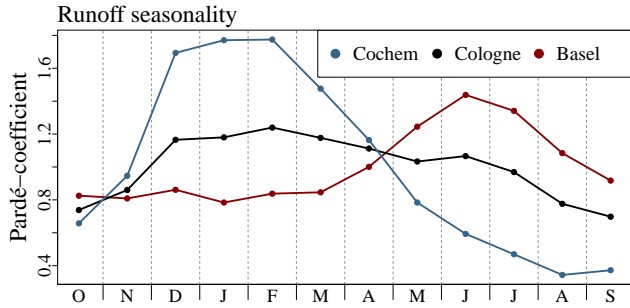

**Figure 4.** Pardé-coefficients (ratio of average montly discharge and the mean annual discharge) (Pardé, 1933; Spreafico and Weingartner, 2005) for gauges Cochem, Basel and Cologne calculated based on measured discharge from the time frame 1971 to 2000.

evapotranspiration need to prevail longer than only a few days. According to our experience, a 10-day window width provides a good estimate to assess potential impacts on streamflow.

In the case of annual maxima, we display the timing and magnitude as boxplots and histograms. The length of the boxplot whiskers is 1.5 times the interquartile range (IQR). However, if no data point exceeds this distance, the whiskers only reach until
the minimum/maximum value. The notches extent to $+/-1.58 \cdot \frac{IQR}{\sqrt{n}}$ with n being the length of the data vector (McGill et al., 1978; R Core Team, 2019). The notches roughly represent 95% confidence intervals for the difference in two medians. For visualisation purposes, we do not display whiskers and outliers of boxplots displaying monthly maxima values. Histograms always depict the probability density and have a total area of one. To estimate the snowmelt contribution with regard to annual streamflow peaks, we calculate the ratio between snowmelt the preceding 10 days and snowmelt the preceding 10 days
plus precipitation the preceding 5 days ($S_{frac}$). Furthermore, we estimate evapotranspiration loss as the ration between actual evapotranspiration the preceding 10 days and snowmelt the preceding 10 days plus precipitation the preceding 5 days ($ET_{loss}$). In addition, we determine mean average annual cycles of $S_{frac}$, the average elevation of the snowmelt ($S_{elev}$) and the solid fraction of precipitation ($P_{solid}$) and the median average annual cycle of $ET_{loss}$.

In the framework of the analysis, we focus on the three gauges: Basel, Cochem and Cologne (Fig. 3). Selected gauges
and sub-basins enable a detailed insight into changes in pluvial and nival processes and changes in the main channel of the Rhine River. Gauge Basel is located at the transition from High to Upper Rhine. The basin upstream gauge Basel encompasses large areas of high alpine character. Snowmelt during spring and early summer is an important runoff/flood-generating process (Wetter et al., 2011; Stahl et al., 2016). Runoff at gauge Cochem (Moselle River) is characterised by a pluvial flow regime with high runoff during winter and low runoff during summer (Fig. 4). Flooding typically occurs in winter and early spring due
to large-scale advective precipitation (Pfister et al., 2004; Bronstert et al., 2007). The gauge Cologne is located in the Lower Rhine region after the confluences of the main tributaries Moselle, Neckar and Main (Fig. 2). Streamflow at gauge Cologne is characterised by a complex flow regime containing both nival and pluvial characteristics.

**Table 2.** Names/Abbreviations, descriptions and units of variables investigated on sub-basin level.

| Variable | Description | Unit |
|---|---|---|
| $P_{max5}$ | 5-day precipitation maxima (total or liquid) | mm |
| $S_{max10}$ | 10-day snowmelt maxima | mm |
| $ET_{max10}$ | 10-day actual evapotranspiration maxima | mm |
| $S_{frac}$ | Contribution of snowmelt to streamflow estimated as the ratio between snowmelt the preceding 10-days and snowmelt the preceding 10 days plus liquid precipitation the preceding 5 days | % |
| $ET_{loss}$ | Evapotranspiration loss estimated as the ratio between actual evapotranspiration the preceding 10-days and snowmelt the preceding 10 days plus liquid precipitation the preceding 5 days | % |
| $S_{elev}$ | Average elevation of snowmelt | m |
| $P_{solid}$ | Solid fraction of precipitation (snowfall) | % |

## 3 Results

### 3.1 Annual maxima

The magnitudes of annual streamflow maxima at gauge Basel increase with rising temperatures (Fig. 5 a). However, this increase is not linear with the magnitude of the warming. The most prominent increase shows up between the historic time frame (1971–2000) and the 1.5 °C warming level. According to the model simulations, the median of annual streamflow maxima increases from 2557 m$^3$ s$^{-1}$ in the historic period to 2827 m$^3$ s$^{-1}$ supposing a warming of 1.5 °C. Among the different warming levels we distinguish marginal differences (Fig. 5 a). At gauge Basel, annual streamflow maxima occur throughout the year (Fig. 5 d). In the historical period, runoff peaks cluster during spring and early summer (snowmelt season). In a warming climate, this cluster is more and more dispersed and annual maxima are increasingly recorded during winter, in particular for the 3 °C warming level. At gauge Cochem, no clear signals of change are detected, neither for the magnitudes nor the timing of annual streamflow maxima (Fig. 5 b and e). At gauge Cologne, streamflow maxima tend to be stronger at the selected warming levels compared to the historic time frame (Fig. 5 c and f). Again, differences among warming levels are small.

For both gauges Basel and Cochem, the estimated contribution of snowmelt to annual streamflow maxima ($S_{frac}$) strongly decreases with rising temperatures (Fig. 6 a and b). At gauge Basel (Cochem), the median of $S_{frac}$ decreases from 15.7% (23.0%) during the historical time frame to 6.7% (0.2%) at a 3 °C warming. At a 3 °C warming, only 27.2% (16.8%) of the annual streamflow maxima have an estimated snowmelt contribution of more than 15% at gauge Basel (Cochem). For both gauges Basel and Cochem, magnitudes of $S_{max10}$ diminish (Fig. 6 c and d). The median of annual $S_{max10}$ for gauge Basel

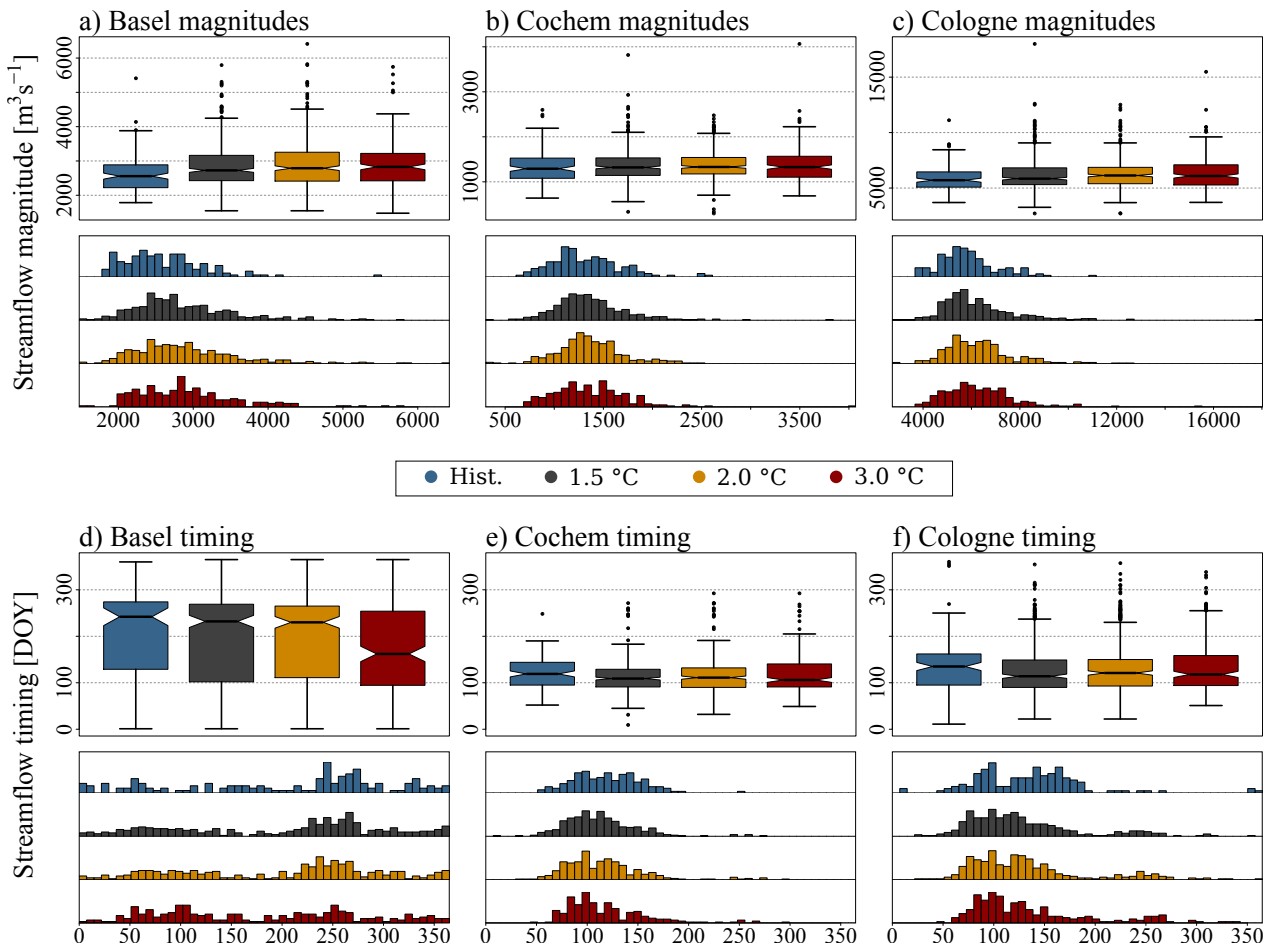

**Figure 5.** Magnitudes and timing (hydrological year starting 1. October) of annual streamflow maxima simulated for gauges Basel, Cochem and Cologne under selected warming levels (14 GCM-RCP realisations reach 1.5 °C, 13 reach 2 °C and 8 reach 3 °C warming) and displayed as boxplots and histograms. Histograms depict probability density and have a total area of one.

(Cochem) is around 32.6 mm (23.9 mm) in the historic time frame and is reduced to 20.6 mm (8.5 mm) at a 3 °C warming. At gauge Basel, $S_{max10}$ do not only get weaker, they also tend to be recorded earlier in the hydrological year (Fig. 6 e). At gauge Cochem, the timing of annual 10-day snowmelt maxima ($S_{max10}$) remains unchanged (Fig. 6 f). In both sub-basin, liquid and total annual $P_{max5}$ increase with rising temperatures (Fig. 6 g, h, i, and j). At gauge Basel (Cochem), the median of liquid

5    annual $P_{max5}$ increases from 63.4 mm (43.9 mm) in the historic time frame to 74.4 mm (50.5 mm) at a 3 °C rise in temperature. The median of the estimated evaporation loss during the genesis of annual streamflow maxima ($ET_{loss}$) is 21.8% (9.2%) at gauge Basel (Cochem) during the historic time period (Fig. 6 k an l). At gauge Basel, $ET_{loss}$ remain fairly stable for moderate warming levels (1.5 and 2 °C) and strongly decreases to 15.4 mm at a 3.0 °C warming, as streamflow peaks more frequently are recorded during winter. At gauge Cochem, the median of $ET_{loss}$ remains almost unchanged and has a value of 9.4% at

a 3 °C warming. Magnitudes of annual $ET_{max10}$ increase with rising temperatures (Fig. 6 m and n). At a 3 °C warming, the median of $ET_{max10}$ magnitudes increases by 11.7% (6.2%) for gauge Basel (Cochem) compared to the historic simulations.

## 3.2 Annual cycles

At gauge Basel (Cochem), the solid fraction of precipitation ($P_{solid}$) reaches values of 69.9 % (43.9 %) during winter in the
historic time frame (Fig.7 a and b). Our results indicate that at a 3 °C warming, on average, the fraction of solid precipitation will be reduced to less than 40 % (17 %) at gauge Basel (Cochem) in winter. At gauge Basel, the estimated average contribution of snowmelt to streamflow ($S_{frac}$) reaches values values up to 40 % during winter, spring and early summer (Fig. 7 c). Strongest decreases in $S_{frac}$ show up in summer (Fig. 7 c). In the Moselle catchment at gauge Cochem, $S_{frac}$ values strongly decrease during the cold season (Fig. 7 d). Upstream of Basel, the average melt elevation ($S_{elev}$) is moving upward the elevation range
throughout the year (Fig. 7 e). On average, $S_{elev}$ is 359 m higher at 3 °C warming compared to the historic time period. At gauge Cochem, $S_{elev}$ is restricted to elevations below 1100 m (Fig. 7 f). Simulation results hint at higher $S_{elev}$, particularly at the beginning and end of the snow season. However, changes are less prominent compared changes detected at gauge Basel. At gauge Basel, the estimated average evapotranspiration loss ($ET_{loss}$) is below 100 % almost throughout the year (Fig. 7 g). Only during summer months and more frequently with stronger warming, $ET_{loss}$ reach values above 100 %. At gauge Cochem,
$ET_{loss}$ are below 100 % between October and March (Fig. 7 h). During the course of the summer, average $ET_{loss}$ can reach values up to almost 400 %.

## 3.3 Monthly maxima

At gauge Basel, monthly streamflow maxima generally increase during winter and decrease in late summer (Fig. 8 a). Streamflow maxima in May and June seem to increase in magnitude at the more moderate warming levels (up to a warming of 2 °C)
and decrease as warming progresses. A similar pattern of initial increases in monthly maxima and a subsequent stabilisation or even a decrease at higher warming levels shows up in December and January at gauge Cochem (Fig. 8 b) and in all winter months at gauge Cologne (Fig. 8 c). In general, patterns of change in monthly streamflow maxima at gauge Cologne seem to reflect an overlap of features visible at gauges Basel and Cochem.

At gauge Basel, magnitudes of $S_{max10}$ remain fairly stable during winter (Fig. 9 a). Strong decreases in $S_{max10}$ show up
in spring and are most pronounced from May to July. In the Moselle catchment at gauge Cochem, $S_{max10}$ strongly decrease throughout the cold season (Fig. 9 b). $P_{max5}$ tend to increase throughout the year (Fig. 9 c, d, e and f). In the Moselle catchment, no big differences between changes in liquid and total $P_{max5}$ are detected. In the Rhine Basin upstream gauge Basel, rising temperatures seem to evoke changes from solid to liquid precipitation, which enhance the overall increase in 5-day precipitation sums, particularly in the cold season (Fig. 9 c and e). Our model simulation suggest that evapotranspiration only plays a minor
role in the Rhine Basin during winter (Fig. 9 g and h). We detect highest values of $ET_{max10}$ reaching up to 35 mm for the sub-basin at Cochem during summer. Values of $ET_{max10}$ increase with rising temperatures.

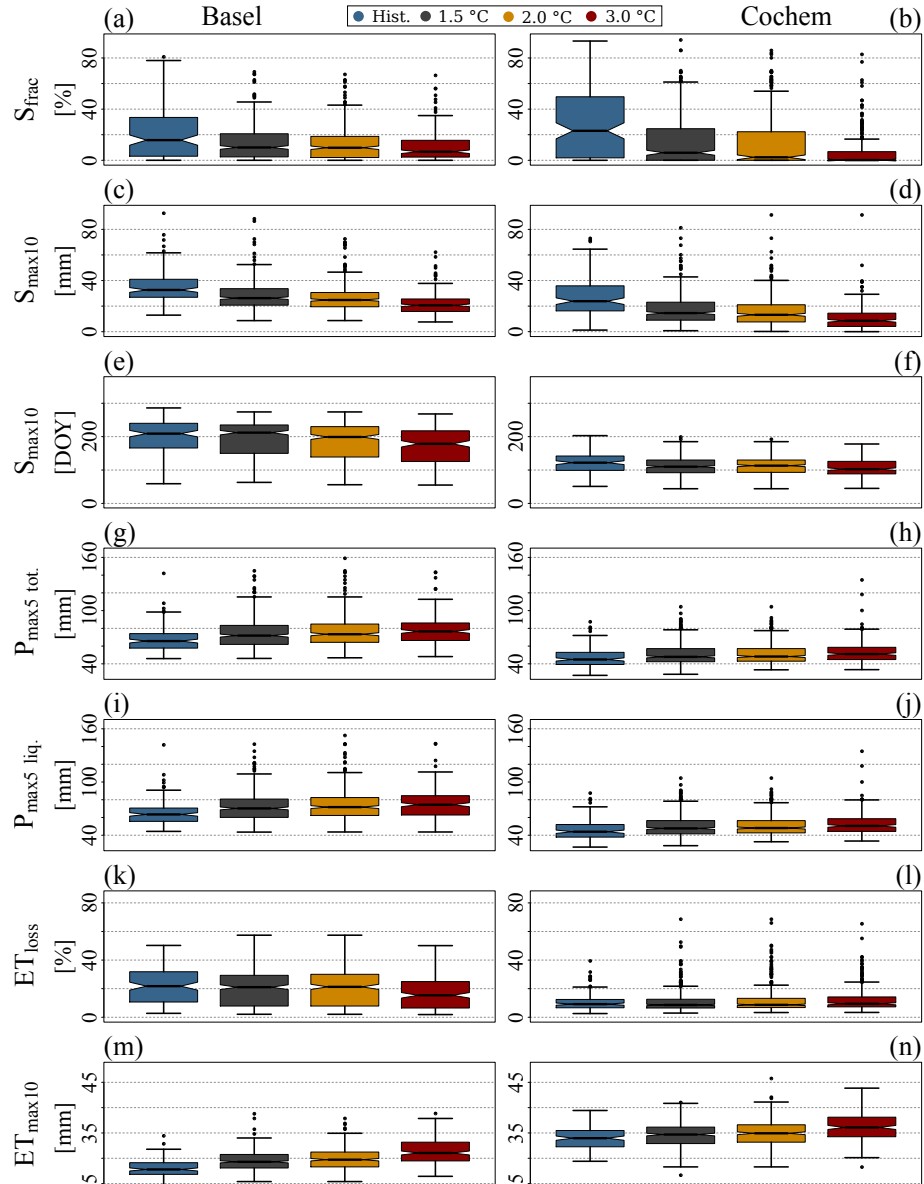

**Figure 6.** Estimated contribution of snowmelt to the annual streamflow maxima ($S_{frac}$; a and b), magnitudes (c and d) and timing (e and f) of annual 10-day snowmelt maxima ($S_{max10}$), magnitudes of annual total (g and h) and liquid (i and j) 5-day precipitation maxima ($P_{max5}$), estimated evapotranspiration loss during the genesis of annual streamflow maxima ($ET_{loss}$; k and l) and magnitudes of annual 10-day actual evapotranspiration maxima ($ET_{max10}$; k and l) for sub-basins at Basel (left column) and Cochem (right column) under selected warming levels (14 GCM-RCP realisations reach 1.5 °C, 13 reach 2 °C and 8 reach 3 °C warming).

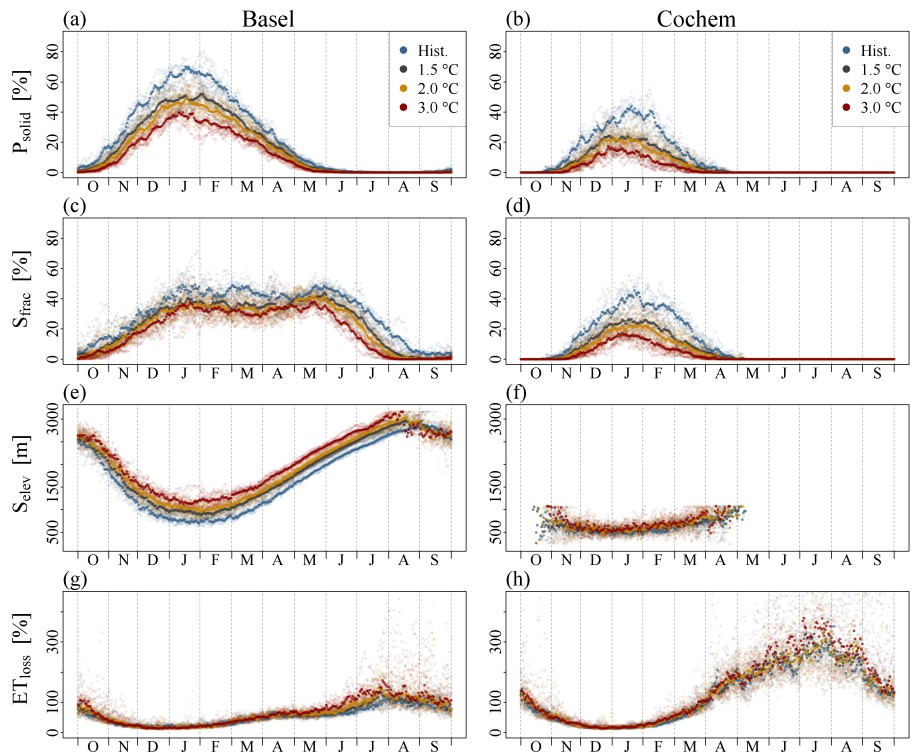

**Figure 7.** Mean annual cycles of the fraction of solid precipitation ($P_{solid}$; a and b), estimated contribution of snowmelt to streamflow ($S_{frac}$; c and d), average elevation of snowmelt ($S_{elev}$; e and f) and estimated evapotranspiration loss ($ET_{loss}$; g and h) for sub-basins at Basel and Cochem under selected warming levels (14 GCM-RCP realisations reach 1.5 °C, 13 reach 2 °C and 8 reach 3 °C warming).

## 4 Discussion

Rising temperatures diminish seasonal snow covers (see also Bavay et al., 2009; Rousselot et al., 2012; Schmucki et al., 2015; Beniston et al., 2018). As a result, the importance of snowmelt as a flood-generating process decreases (Fig. 6 a, b, c and d). In the Rhine Basin at Basel, 10-day snowmelt maxima ($S_{max10}$) decrease for all months of spring and summer (Fig. 8 a). At

5  no point in time during the snowmelt season, a warming climate results in an increase in risk of snowmelt-driven flooding. Our results indicate that the detected earlier timing of the annual snowmelt maxima (Fig. 6 e) is not due to an increase in snowmelt magnitudes earlier in the year. It rather seems that events early in the snowmelt season, even if weakened by rising temperatures, more often are the strongest of the year already, as snow packs are increasingly depleted within the course of the snowmelt season. For the basin at Basel, we can not find indications that an earlier snowmelt due to rising temperatures

10  shifts the risk of snowmelt-driven flooding forward in time. Despite the temporal shift forward of annual snowmelt maxima, flood hazard seems to decrease, as the temporal shift concurs with a strong decrease in snowmelt magnitudes (Fig. 6 c). Our findings go along with results from Musselman et al. (2017), who suggest that a "shallower snowpack melts earlier, and at lower rates, than deeper, later-lying snow-cover". However, the disappearance of snow packs and glaciers is likely to favour

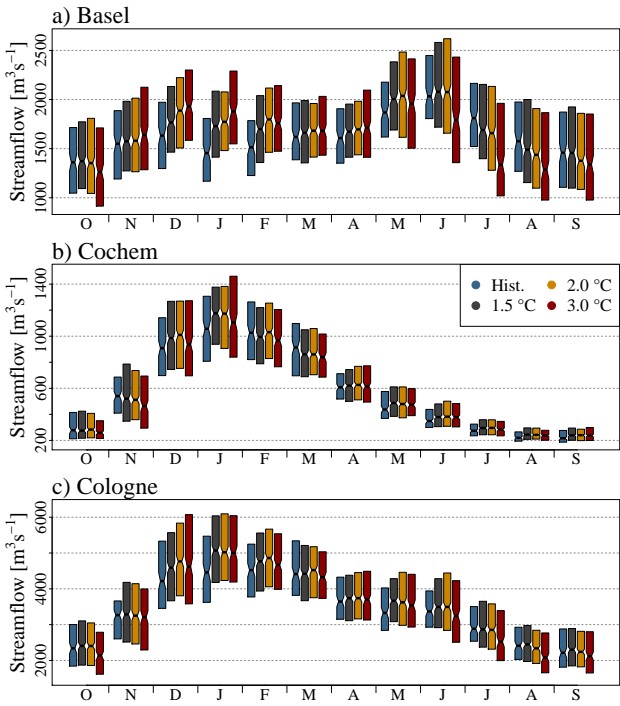

**Figure 8.** Magnitudes of monthly streamflow maxima simulated for gauges a) Basel, b) Cochem and c) Cologne under selected warming levels (14 GCM-RCP realisations reach 1.5 °C, 13 reach 2 °C and 8 reach 3 °C warming). Whiskers and outliers of the boxplots are not displayed.

low-flow conditions along the Rhine River (Junghans et al., 2011; Stahl et al., 2016). Another factor having the potential to initiate or reinforce low-flow situation are increasing values of evapotranspiration, particularly during summer (Fig. 9 g and h).

Our results indicate that at Basel during winter, the lack of snowmelt from lower elevations, at least partly, is compensated by snowmelt from areas located at higher elevations (Rottler et al., 2021) (Fig. 7 c and e and Fig. 9 a). This compensation effect

5   seems to be increasingly insufficient as the snowmelt season progresses and the snowline moves upward. We suggest that in winter, the almost unchanged potential of snowmelt-induced runoff at Basel encounters increased antecedent precipitation (Fig. 9 c), in turn, resulting in a strong increase in streamflow maxima (Fig. 8 a). Our results confirm previous studies suggesting that rising temperatures might lead to stronger precipitation events (e.g., Lehmann et al., 2015; Alfieri et al., 2015; King and Karoly, 2017; Bürger et al., 2019; Rottler et al., 2020) (Fig. 6 g-j and Fig. 9 c-f) and a shift from solid to liquid rainfall (e.g.,

10   Allamano et al., 2009; Addor et al., 2014; Davenport et al., 2020) (Fig. 7 a and b). In catchments having mixed hydrological regimes with rainfall and snowmelt, rising temperatures seem to lead to a shift from snowmelt to rainfall as most important flood generating process (Vormoor et al., 2015, 2016). Reconstructing the largest floods in the High Rhine since 1268, Wetter et al. (2011) indicate that about half of all large events occurred during summer due heavy precipitation combined with high

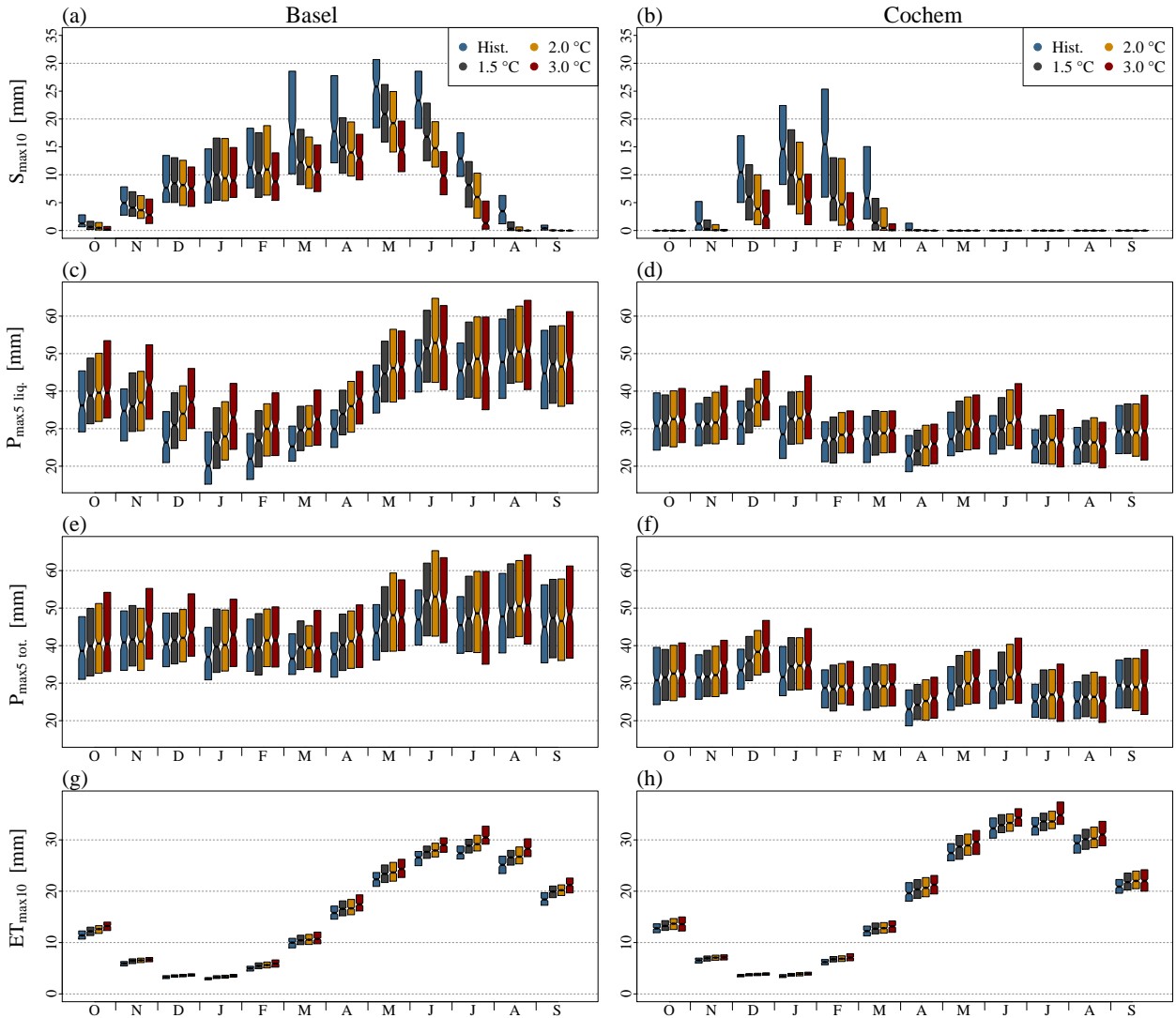

**Figure 9.** Magnitudes of 10-day snowmelt maxima ($S_{max10}$; a and b), liquid (c and d) and total (e and f) 5-day precipitation ($P_{max5}$) and 10-day actual evapotranspiration maxima ($ET_{max10}$; g and h) for sub-basins at Basel and Cochem under selected warming levels (14 GCM-RCP realisations reach 1.5 °C, 13 reach 2 °C and 8 reach 3 °C warming). Whiskers and outliers of the boxplots are not displayed.

baseflow from snow- and ice-melt. Our results indicate that with rising temperatures, most flood events will occur in winter (Fig. 5 d).

In March and April, the increase in rainfall amounts in the basin at Basel compares to increases in winter, the magnitudes of streamflow maxima, however, hardly change (Fig. 8 a). We suggest that the increasing potential of rainfall-induced flooding 5 is counterbalanced by decreasing snowmelt (Fig. 9 a and c). Furthermore, our results hint at a transient increase in flood

magnitudes during May and June (Fig. 8 a). It seems that during those two months, snowmelt is still strong enough to support an increase in streamflow peaks due to increased antecedent precipitation at moderate warming levels (1.5 °C and 2.0 °C). With further rising temperatures, however, the magnitudes of streamflow maxima reduce along with declining snowmelt (Fig. 8 a). The mHM model set-up that we use to simulate the Rhine River does not include a lake module. The simulation results attained for the Rhine Basin, particularly for gauge Basel, can be further refined by the representation of the large lakes located in Switzerland and Southern Germany (Imhoff et al., 2020). The large storage volume and the possibility to regulate lake levels dampen streamflow peaks.

For gauge Cochem and the associated sub-basin of the Moselle River, we detect similar counterbalancing effects between snowmelt and rainfall: an increasing flood potential due to increased precipitation amounts encounters declining snow packs. Again, decreases in snowmelt magnitudes seem to counterbalance increased precipitation resulting in comparatively small and transient increases in streamflow maxima (Fig. 8 b and Fig. 9 b and d). As highest mountains in the sub-basin only reach up to around 1300 m a.s.l., snowmelt compensation effects, i.e., snowmelt from higher elevations, at least partly, replaces the lack of snowmelt from lower elevation, only plays a marginal role. Analysing changes in frequencies of rain-on-snow (RoS) events with flood-generating potential for large parts of Europe for the historic time frame 1950–2011, Freudiger et al. (2014) hint at similar processes changing flood hazard. Their analyses suggest an increase in flood hazard from RoS events in medium-elevation mountain ranges in the Rhine River Basin in winter due to increased rainfall and a decrease in RoS events in spring due to decreases in snow cover. Although important Rhine tributaries, such as the Moselle River, often are characterised as pluvial-type rivers, the importance of snowmelt as runoff component must not be underestimated. Simulating the Rhine River for the time frame 1901–2006, Stahl et al. (2016) suggest that at gauge Cochem, 26 % of the annual streamflow originates from snowmelt. During winter, this fraction increases up to almost 40 % (see also Fig. 7 b). However, the inter-annual variability of annual streamflow and the relative fractions of streamflow components is high, particularly in pluvial-type tributaries of the Rhine River (Stahl et al., 2016).

In Cologne, which is located at the main stream after the confluence of all major tributaries, signals emerging from the different sub-basin superimpose. Accordingly, we detect increases in runoff peaks during winter (Fig. 8 c). Detected increases seem to level off as temperatures continue to rise beyond the 2 °C warming level. We do not find indications supporting the hypothesis describing the creation of a new flood type in the Rhine River Basin due to a transient merging of nival and pluvial flood types. We detect counterbalancing effects between changes in snowmelt and precipitation within the sub-basins. Rising temperatures strongly reduce snowfall, snow accumulation and the snow volume available for melt. The reduction in snowmelt-driven runoff during flood genesis seems to impede the increase in streamflow peaks due to increases in antecedent precipitation. Caution has to be exercised labelling basins such as the Moselle catchment as pluvial-type or the Rhine Basin at Basel as nival-type. In both sub-basins, snowmelt and precipitation are important factors for flood generation. In the framework of this study, we mostly focus on changes in streamflow seasonality and analyse average changes in streamflow generating mechanisms. A detailed analysis of isolated extremes simulated is still pending.

# 5    Conclusions

We investigate changes in flood seasonality in the Rhine River Basin under 1.5, 2.0 and 3.0 °C warming using the spatially distributed hydrologic model mHM. In order to improve our understanding of changes in rainfall- and snowmelt-driven runoff, we carried out a detailed inspection of the Rhine River Basin at Basel and the Moselle River Basin at Cochem. We detect significant changes in both rainfall- and snowmelt-driven runoff peaks. Rising temperatures deplete seasonal snowpacks. As a consequence, the importance of snowmelt as flood-generating process diminishes. At no time during the year, a warming climate results in an increase in the risk of snowmelt-driven flooding. Furthermore, solid precipitation (snowfall) strongly decreases during winter. The shift from solid to liquid precipitation further enhances the overall increase in antecedent precipitation.

Our results indicate, that in order to understand changes in annual and monthly streamflow maxima, the examination of counterbalancing effects between changes in snowmelt- and rainfall-driven runoff is crucial. We suggest that future changes in flood characteristics in the Rhine River Basin are controlled by increased precipitation amounts on the one hand, and reduced snowmelt on the other hand. The nature of their interaction defines the type of change in runoff peaks. In the case of the Moselle River, increased rainfall amounts during winter, at least partly, are counterbalanced by reduced snowmelt contribution to the streamflow peaks, resulting in only small or transient changes. In the Rhine Basin at Basel, strong increases in antecedent liquid precipitation encounter almost unchanged snowmelt-driven runoff during winter. Hence, streamflow maxima increase strongly. During May and June, our results hint at a transient increase in streamflow magnitudes at gauge Basel (Fig. 8 a). It seems that snowmelt is still strong enough to support an increase in streamflow peaks due to increased antecedent precipitation at moderate warming levels (1.5 °C and 2.0 °C). With further rising temperatures, however, the magnitudes of streamflow maxima reduce along with declining snowmelt (Fig. 8 a). In addition to a strong decline in snow packs in the Alps, we detect an upward movement of the snowmelt elevation. It seems that during winter, snowmelt from higher elevation, at least partly, can replace snowmelt for elevations below (Rottler et al., 2021). Our findings confirm previous investigations suggesting a shift from snowmelt to precipitation as most important flood generating mechanism (Vormoor et al., 2015, 2016). We can not find indications of a transient merging of pluvial and nival flood types in the Rhine Basin.

The understanding of future changes in flood characteristics along the Rhine River and its tributaries is of great importance for water resources and flood management. Within this study, some progress has been made in assessing the importance of rainfall and snowmelt as flood-generating processes under different warming levels. However, only further studies pursuing the improvement of meteorological input data and hydrological modelling can ensure a comprehensive understanding of future flood characteristics in the Rhine River. Next steps could be the implementation and validation of a physically-based snow routine and a glacier module in mHM in order to substantiate our current results regarding the relevance of snowmelt magnitude and timing for the generation of Rhine floods. The usage of satellite-based snow cover maps during model calibration and/or validation might further improve the simulation of the snow cover dynamics. A streamflow component model enabling the tracing of river flow originating processes (e.g., Stahl et al., 2016) might ameliorate the understanding of snowmelt and rainfall

as flood-generating processes at different Rhine gauges. Furthermore, the representation of lakes (e.g., Imhoff et al., 2020) and reservoirs and their management might improve streamflow simulations, particularly during low-flow conditions.

*Code and data availability.* Source code of the hydrologic model mHM v.5.10 can be accessed at https://git.ufz.de/mhm/mhm (last access: 8 October 2020). R-scripts used to analyse simulation results are available at https://github.com/ERottler/mhm_rhine (last access: 9 November 2020). Discharge data can be requested from the Global Runoff Data Centre, 56068, Koblenz, Germany (GRDC). Further data sets used can be made available upon request.

*Author contributions.* ER conducted the analysis and wrote the manuscript. AB, GB and OR provided support and guidance in the process of model set up, data analysis and preparation of the manuscript.

*Competing interests.* The authors declare that they have no conflict of interest.

*Acknowledgements.* This research was supported by the Deutsche Forschungsgemeinschaft (GRK 2043/1-P2) within the NatRiskChange research training group at the University of Potsdam (https://www.uni-potsdam.de/natriskchange/, last access: 2 October 2020). We acknowledge the datasets generated in the EDgE proof-of-concept project performed under a contract for the Copernicus Climate Change Service (http://edge.climate.copernicus.eu, last access: 8 October 2020). ECMWF implements this service and the Copernicus Atmosphere Monitoring Service on behalf of the European Commission. We acknowledge EDgE colleagues Rohini Kumar and Stephan Thober for establishing the mHM model setup and performing the downscaling of the CMIP5 data sets, respectively. We acknowledge the E-OBS dataset from the EU FP6 project ENSEMBLES (http://ensembles-eu.metoffice.com) and the data providers in the ECA&D project (http://www.ecad.eu). We acknowledge the ISI-MIP project for providing the bias corrected CMIP5 climate model data. The Copernicus Land Monitoring Service, implemented by the European Environmental Agency, provided the European Digital Elevation Model (EU-DEM), version 1.1. We also acknowledge the HOKLIM project (https://www.ufz.de/hoklim) by the German Ministry for Education and Research (grant number 01LS1611A). We also thank various other organisations and projects for providing data used in this study, including JRC, ESA, NASA, USGS, GRDC, BGR, UNESCO, ISRIC, and EEA.

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

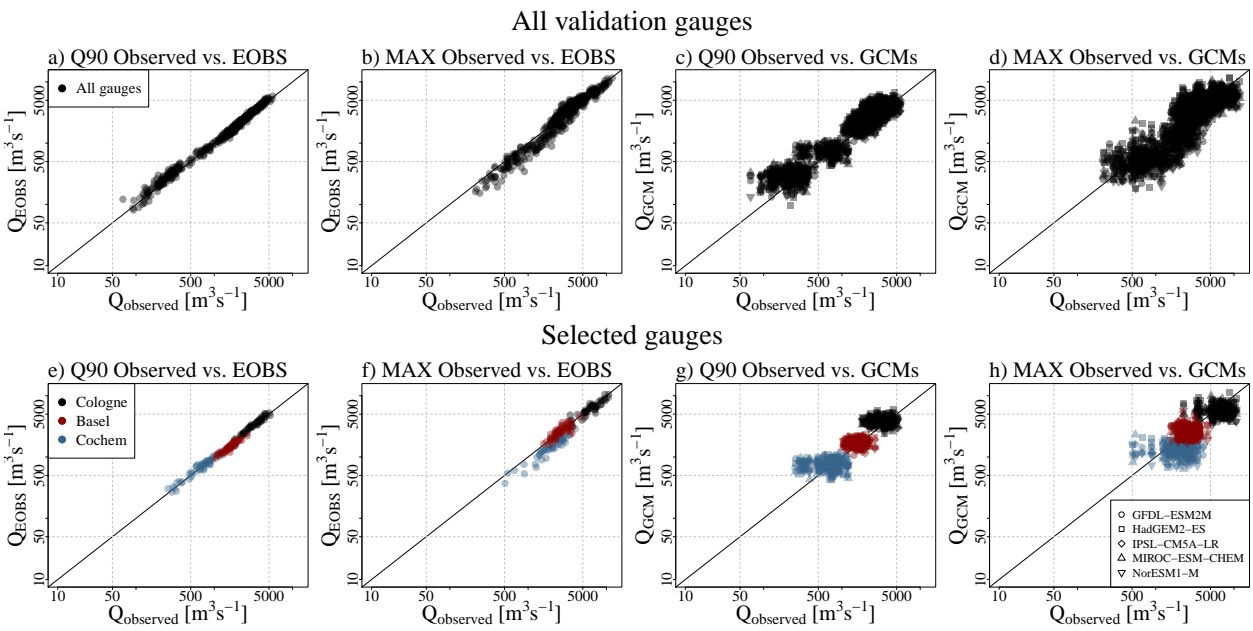

**Figure A1.** Scatter plot of observed and simulated annual streamflow maxima (MAX) and the 90 % streamflow quantile (Q90) of the hydrological year starting 1 October for all validation gauges (a-d; Fig. 2) and for selected gauges (e-h). Panels a, b, e and f depict observed discharge and simulated discharge using E-OBS-based meteorological forcing. Panels c, d, g and h depict observed discharge and simulated discharge using climate model data from the ISI-MIP project. Time frame investigated: 1951–2000.

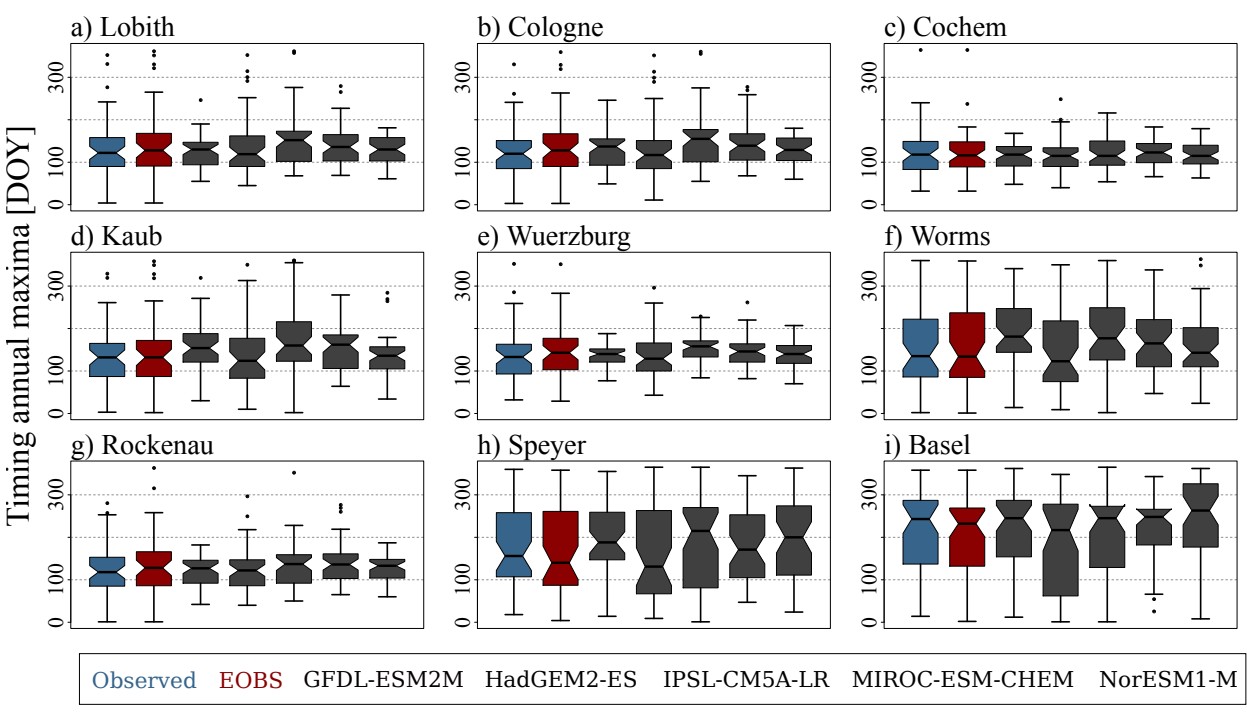

**Figure B1.** Timing of annual streamflow maxima observed and simulated using E-OBS-based meteorological forcing and climate model data from the ISI-MIP project for all validation gauges (Fig. 2). Time frame investigated: 1951–2000.

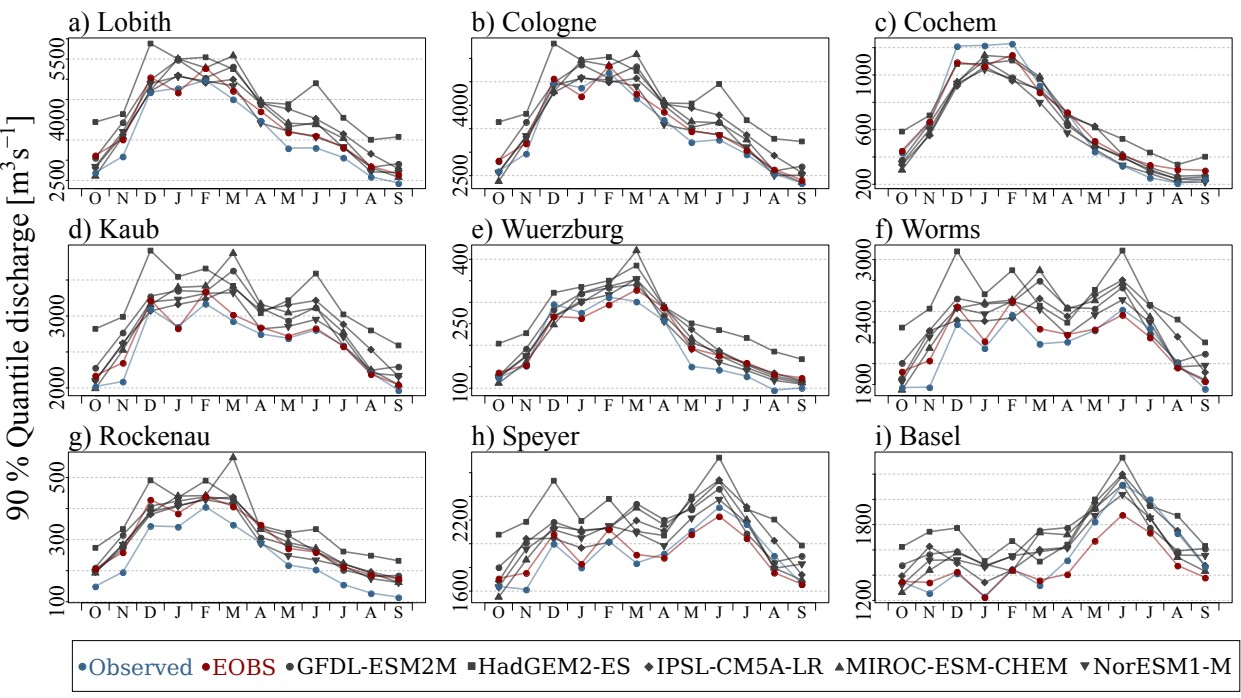

**Figure C1.** Streamflow quantiles (90 %) for every month of the year based on daily resolution observations and simulations using E-OBS-based meteorological forcing and climate model data from the ISI-MIP project for all validation gauges (Fig. 2). Time frame investigated: 1951–2000.