# Peer review of "Projected changes in Rhine River flood seasonality under global warming"

_Hydrology and Earth System Sciences, 2020_

## Referee Comment (RC1) · Anonymous Referee #1 · 20 Dec 2020

The manuscript is much relevant both for science and practical water ressources management. It helps to understand the complexe runoff regime of the river Rhine and the effects of global warming on flood generation in large rivers. Furthermore it is a very helpful study for the planning of flood protection measures.

The title reflects the content of the paper very well. The results are sufficient to support the interpretations and the clearly presented conclusions. The figures are very clear and for the most part self-explanatory.

General comment:

Scientific terms and abbreviations should be the same in figures, tables and the text. E.g. in Fig 5 the term "discharge" is used but in the text "streamflow". in Fig 6 and

its captions is written "melt magnitude" but in table 2 and the text the abbreviation Smax14. In table 2 Smax14 is singular, in the text sometimes plural. In the figures, tables and their captions should be checked if the same terms are used as in the text. I suggest to give the explanation of abbreviations in the method chapter, in table 2 and in the captions of the figures (but there only in brackets).

Data and Methods:

The expression "sub-basin upstream gauge Basel" is quite complicate and, as I think, not necessary. I suggest to explain, that some of the investigated variables refer to the gauge and others to the basin. And then I would refer only to "gauge Basel" or "basin of gauge Basel"). Same for Cochem. Especially in the later chapters the long term of "sub-basin upstream gauge Basel" is a bit confusing.

P7, Fig3: Missspelling in the last point (elevation and solid precipitation)

P8, Table 2: the two last variables are not listed on P6, L11-L13. For me it is not clear what is meant with "melt elevation" especially when it comes to the units (see also Fig 6). For more clearity one could give the units in the table that also helps to understand if it is a value at a gauge or for a basin.

Results:

It would be helpful, if the figures would be described more systematically. Sometimes exact values are given, sometimes not. Sometimes the results for Basel are described first, sometimes those for Cochem. It would be less confusing for the reader, wenn the order would be always like in the figures.

P8, L1: In my opinion the first two sentences contain already important results and therefore should be more precise and with more information. E.g. like this: "According to the model simulations the changes are largest at gauge Basel (Fig. 5a). Here, the median of discharge magnitude increases from 2500(?) $m^3$/s in the historic period to 2700 (?) $m^3$/s supposing a warming of 1.5 degree. Furthermore, the highest floods are

higher than in the historic period. However, this increase in discharge is not linear with temperature rise....".

P8, L4: The results for Cochem and Cologne (Fig 5b and 5c) should be briefly mentioned already here (e.g. Cochem shows only very little increase of maximum discharge with increasing temperature).

P8, L5: The figure reference is not correct. Should be Fig. 5d to 5f.

P8, L8: I see a signal of change for Cochem: annual maxima seems to be a bit earlier (probably due to less solid fraction in the Vosges as prooved in fig 6) .

P8, L13: Wrong word? "...runoff contribution of snowmelt of more the 20%..." should probably be "...runoff contribution of snowmelt of more than 20%...".

P8, L15: Here, Smax14 is plural and in the sentence before singular.

P8, L16: "solid Pmax5" is probably wrong, "total Pmax5" is probably meant as shown in the Figure and als logically (solid Pmax5 decreases with temperature rise).

P8, L22: Value of solid fraction should also be given for the historic period (80%), so that the reader do not have to look for it in the figure.

P11, L1: In think the information on highest ETmax14 is not relevant for the purpose of this study that is flood seasonality. This sentence should be deleted.

Discussion and Conclusion:

P12, L3: Delete sixth word "in".

P14, L32: Concerning precipitation intensity (see also Abstract): I think the study does not show, that the precipitation intensity increases. The rainfall intensity (or I would say "rainfall amount" because I associate intensity with shorter event of up to 72 h) increases due to higher fraction of liquid precipitation. Less snow, more rain... But is the total amount of precipitation increasing? If yes, I missed this result before.
P15, L14: Here one could mention the lake of Constance, that has a considerable influence on the flood magnitude at gauge Basel. The lake of Constance is a big storage for the snow melt from the "Alpenrhein". By the way, the lake is not shown on the maps (and I wonder if it is part of the mHM). At least one should briefly mention the lake and its effect in general.

———————————————————

---

## Referee Comment (RC2) · Anonymous Referee #2 · 23 Dec 2020

**1. General comments**

Overall quality of the preprint: A well-structured paper that supports earlier results and adds additional insights into the shifts of flood genesis under climate change. The latter could be highlighted a bit more in the abstract and other parts of the text (suggestions under "specific comments). Principle review criteria (scientific significance, scientific quality, and presentation quality) are generally evaluated as "good". Suggest to accept with revisions.

**2. Specific comments**

The specific individual scientific questions/issues are labelled as "Change requests" that should be changed before publication and "Suggestions" that could be changed

before publication.

- Page 1, Abstract - change request: The abstract describes basic mechanisms of the flow regime of the Rhine River in a warmer climate. This is neither new – cf. e.g. to Kwadijk Romans (1995; https://link.springer.com/article/10.1007/BF01093854) - nor the core of the study presented here. It is suggested (a) to highlight a set of change signals of the hydrological characteristics you evaluated (number of years with snowmelt fraction above a threshold or lift of "melt elevation" etc.) and/or (b) focus more on the hypothesized new flood type superimposing rainfall- und snowmelt-induced runoff.

- Page 1, line 16 - suggestion: The term "current climate crisis" has a political flavor.

- Page 2, lines 1ff - suggestion: Add reference to IPCC SROCC

- Page 3, lines 9ff - change request: Please add here, that hydrological processes are modelled at 5 km grid resolution (referring to page 5). Otherwise the reader waits for a final downscaling step of met. data to the 500 m grid of mHM.

- Page 3, line 16 – change request: The quoting used here reads like the authors do not understand what this part of the procedure/sentence means. Is that the intention here? The bias correction procedure is important when dealing with peak flow analyses (and heavy precipitation). Please rephrase.

- Page 3, line 22ff – change request: Please explain how you treated the catchment upstream of Basel. As it reads now you would end up with two parameter sets; one from the calibration of Basel, one from the calibration of Lobith (also containing the catchment upstream of Basel). Please clarify, which parameter set you used for the overlapping part of the catchment or if you used individual model set ups for each gauging station.

[Figure]

- Page 6, line 17f – change request: Please add some details on your experience concerning the 14-day time window for snowmelt and evaporation. Is it based on investigations of historical floods? Or on model simulations?

- Page 6, line 25f – change request: The flow regime at gauge Cologne is usually regarded as "complex" regime containing "nival" and "pluvial" characteristics. This should be added here. Now, the gauge is described as another pluvial example.

- Page 7, Figure 3 – change request: The map shows the Rhine River basin up to Lobith, not the entire Basin. This should be added in the scheme and/or caption.

- Page 7, Figure 4 – suggestion: For reasons of consistency it is suggested not to introduce an additional reference period here (1971-2016). The period 1971-2000 should be chosen here as well.

- Page 8, line 11ff – suggestion: For some readers it may be interesting to note that according to your results there will still be some snowmelt at gauge Cochem even in a $3°C$ warmer world. Suggest to add this point.

- Page 8, line 13 – suggestion: The units of the variables could be changed to give a better "grip" of the results. For example, "the number of streamflow maxima having an estimated runoff contribution of snowmelt of more than 20

- Page 8, line 21 – change request: "Decreases in solid precipitation are most prominent in winter" ← That's not surprising because according to your results the historical period shows is no solid precipitation in summer. Rephrase, e.g. referring to meteorological seasons (DJF, MAM, JJA, SON).

- Page 9, lines 3 – suggestion: Suggest to repeat here that the timing of the highest annual flow remains unchanged.
- Page 10, lines 6f – suggestion: The role of evaporation simulated under climate change conditions strongly depends on the evaporation approach used and the area of interest. Suggest to transport this uncertainty of hydrological modelling by formulating more carefully. For example: "With the approach used here, evaporation seems to play a minor role . . .".

- Page 11, line 6f. – suggestion: It would be also interesting to state already here that snowmelt-driven flooding is possible despite of rising temperatures. At least in low warming levels there may still be relevant snowmelt events. This follows only two pages later.

- Page 11, line 11 – change request: The hypothesis mentioned in the introduction was on flood risks resulting from the overlap of nival and pluvial peak flows. Here, the focus is on snow-melt driven floods only. Check consistency.

- Page 12, figure 8 – suggestion: Add the range that is displayed by the boxes.

- Page 12, line 1f – suggestion: Suggest to stay focused on floods and skip low flows.

- Page 12, line 3ff – change request: In this paragraph it is advisable to be very clear about (a) the statistics (e.g. to avoid confusion between monthly and annual stream flow maxima) and (b) the gauge/regime that is discussed. Otherwise the reader will be lost. For example the statement that "with rising temperatures, most flood events will occur in winter" does obviously not relate to Basel/nival regimes. This has to be more transparent.

- Page 14, line 7 – change request: In how far are peak elevations (here: 1300 m a.s.l.) and the related processes interpreted here reflected in the hydrological model, given the 5 km grid resolution? Please add this to the method description (page 3f.).

- Page 14, line 30ff. – suggestion: cf. comment on abstract. Suggest to refocus this paragraph in the same way.

- Page 15, line 14f. – suggestion: Suggest to stay focused on floods and skip low flows. Lakes and reservoirs play an important role for high flow, too. If they are not yet implemented in the model, this should be mentioned in the methods chapter (page 3f.)

**2. Technical corrections**

- General comment: It was difficult to print the pdf. Presumably one of the graphs is oversampled – please check.

- Page 8, line 5 – change request: Figure 5b contains no information on the timing of runoff maxima. Wrong reference. Please correct (-> 5d?) and repeat the reference to "Basel" in the text/line 5.

- Page 8, line 13 – change request: "more the" -> "more than"

- Page 9, Figure 5 – change request: The horizontal grid lines do only occasionally match the tick marks. Please correct.

- Page 10, Figure 6 – change request: The horizontal grid lines do only occasionally match the tick marks. Please correct.

- Page 12, line 6 – change request: Replace "Smax14" by plain text.

- Page 12, Figure 8 – change request: The horizontal grid lines do only occasionally match the tick marks. Please correct.

- Page 13, Figure 9 – change request: The horizontal grid lines do only occasionally match the tick marks. Please correct.

- Page 24, Figure B1 – change request: The horizontal grid lines do only occasionally match the tick marks. Please correct.

- Page 25, Figure C1 – change request: The horizontal grid lines do only occasionally match the tick marks. Please correct.
* * *

---

## Referee Comment (RC3) · Anonymous Referee #3 · 23 Dec 2020

**1. General**

This paper analyses future changes in flood seasonality in the Rhine River Basin at three different global warming levels using the mesoscale Hydrological Model (mHM). The paper is well structured and written, considers earlier work quite well, and provides new insights in flood seasonality changes under climate change for the Rhine basin. Finally, the authors list some next steps to improve the modelling approach as including a glacier module or reservoir and lake functionality.

**2. Specific comments**

Data and Methods: suggest to include that the model does not include a glacier and lakes module. For the basin upstream Basel not including lakes can have quite some

effect. Now, this becomes only clear at the end of the Conclusions section.

Page 3, line 15, please describe the downscale and bias correction in more detail. The sentence "adjusts the monthly mean and daily variability of simulated climate data to observations." does not describe how this was done.

Page 3, lines 22-25, The calibration procedure could be described in more detail. 1) For example why was the gauge Lobith also included in the calibration procedure? With MPR, one could have choosen for example three smaller sub-basins to find how well parameters are transferable to the larger basin scale. This makes the calibration more efficient, and would also provide an interesting result (although I understand this is not the focus of the paper, it is an important aspect of this study). 2) What were the specific DDS settings (e.g. number of function evaluations)? Please add these to the text. 3) Finally, how many model parameters were calibrated? At least this gives the reader some insight into the model complexity.

Page 14, line 5. Suggest to change "increased precipitation intensity" to amount, the analysis is about a monthly time scale, so probably better to use amount and not intensity.

Page 15, line 10 and lines 13-14: Please add this reference as an example: https://agupubs.onlinelibrary.wiley.com/doi/abs/10.1029/2019WR026807, for a modelling approach also applied to the Rhine basin that already includes a glacier and lake module.

3. Technical corrections

Fig 7. change "elvation" to "elevation"

Page 8, Table 2, change "ration" to "ratio"

---

## Referee Comment (RC4) · Anonymous Referee #4 · 4 Jan 2021

General

The paper addresses a highly relevant subject on changes in flood seasonality in the Rhine basin, by analysing how climate change affects different components of the water balance and their aggregated effect on peak flows. The authors elegantly demonstrate by means of model simulations for different GCMs and climate scenarios the contributions of snow melt and rainfall-driven runoff to peak flow generation over the seasons. These analyses form a relevant contribution to earlier studies on the impacts of climate change on peak flows in the Rhine basin, provide nice insight in the underlying contributions of rainfall and snowmelt, and indicate their effects on time shifts in peak flow occurrences. The paper's title well covers the contents; the paper is well-structured, clearly presents its results in text and figures, and the interpretations, dis-

cussion and conclusions are well supported by the results. I would rate the significance and quality of the paper 'Good'.

Specific comments:

Fig 1: I would suppose that the nival peak not only shifts to earlier in the season but also becomes smaller under CC - as there will be less total snow accumulation over the winter season.

Line 30: 1.5, 2 and 3 degrees warming - relative to 1970 - 2000

Section 2: provide a bit more information on: which parameters did you calibrate? In particular you have a detailed representation of crops and soil types, did you use reference values in all cases, or did you do any calibration here? How did you choose LAI values for different vegetation types and seasons and latitude? How did you perform the bias correction to GCMs for future climates?

P4, L11: assess -> assesses

P4, L13: bases -> is based

P6, L6-9. Do I understand here that the projection times of the periods where the 'targeted' warming was reached was different for each realisation? And with different RCPs you may reach the same warming at different moments (e.g. 1.5 degree under RCP 8.0 early in the century, and RCP2.6 only late) - but to what extent are these scenarios different in your simulations (associated P?). Can you indicate which are the according time horizons used in your simulations?

P6, L14: for the Rhine basin as a whole (e.g. Cologne) a 5 day-period for the precipitation sum seems quite short to generate extreme floods, in particular in view of saturating the soil and travel time of peaks from tributaries.

P6, L19: river discharge at Basel is considerably dampened by the effects of the Swiss lakes. For that reason, earlier studies focused on catchments upstream of the lakes

[Figure]

(e.g. Murg, Thur). Can you indicate to what extent timing and maxima of small peaks - in particular after a dry period with low lake levels - are affected by this?

Figure 5: Please indicate on how many runs each histogram is based. From the methods I read how many GCMs reached each warming, with only 8 of them reaching 3 degrees, but this is not clear for the other histograms. To what extent do different numbers of realisation result in different occurrences of highest extremes, and did different GCMs result in different extremes under - in spite of bias correction?

Fig 5; P8, L9: whereas both for Basel and Cochem there is a decline in the timing of summer maxima for higher temperatures, Cologne shows a small peak emerging around DOY 250 - can you explain this? Consider using the same horizontal scale for figs 5d-f.

P10, L4: is detected -> are detected

P10, fig 6 (k,l): by displaying annual maxima distributions we indeed can see how these shift over time, but we cannot see how shifts evapotranspiration maxima link to peak flows, as the connection to the flood events is lost: we cannot see how much was the 'reduction' of the annual peak flow maxima due to evapotranspiration loss (as you do in fig 6 ab indicating the 'contribution' of snowmelt to the annual maxima). It makes sense that under a warming climate annual maximum evapotranspiration goes up - but if that happens in summer when floods never arise it is hard to judge the role of evapotranspiration in changing peak flows. In fig 9g we can see that the contribution of evapotranspiration change is small indeed.

P11: Discussions -> discussion

P11, L4: ....diminish seasonal snow covers -> please add in a few words the key aspects of that: the total volume, the duration and the timing of melt.

P11, L6: Smax14 is singular.

P11, L8: 'forward' -> in first time use, explicitly explain that you mean: 'earlier in the

year'

P11, L10, 13-14: two factors may play a role: the timing of melting, and the amount of snow that has accumulated so far to be available for melting - you do not indicate the maximum amount of snow that has accumulated by the end of the season to be available for melt.

P11, L15. I do not see a contradiction suggested by using 'however'. Actually, you change subject here to low flow situations, as caused by disappearing glaciers and intensified evapotranspiration, which becomes different from 'lower maxima',

P12. l6 (and in the rest of the paper, in particular in the conclusion, P14, L32): I am not sure whether you should formulate this as 'intense rainfall events' in mm per hour and use this formulation for both summer and winter. 'High intensity' rather relates to high intensity summer storms - as you indicate in lines 12-15 here, but for the winter season I would not formulate that as intensity. Moreover, you consider in your study accumulated precipitation sums over 5 days - that is an amount, not an intensity.

P12, L12. Here you make a relevant statement - that the summer extremes were still supported by snow melt from the Alps to produce their maxima - I presume that that has been derived from the associated historic descriptions. With reduced snow melt in late spring this would indeed reduce the risk of summer floods. Conversely, higher temperatures under future climate change may lead to more intense summer precipitation - still causing higher peak flows. Here we may encounter the questions: does the latter only hold for smaller catchments within the basin, or do these still feed large floods to Cologne? And some GCMs show a N-S difference in precipitation change (some even the signal) across the Rhine basin: is that the case in your experiments? From the histograms in fig 6g this is hard to derive. It would affect probabilities of extreme summer floods, as we experienced in early 2000s in the Elbe.

P14, L3: It is not a true 'interaction' between snow fall and precipitation, but their effects are counterbalancing - as you explain in the following lines.

P14, L9: hint -> suggest

P14, L14: originate - > originates; During this period, we have experienced already over 0.5 degree of warming, so it would be interesting to know what the average for the past few decades would be.

P14, L33: 'intense precipitation events' (see earlier comment): would you describe the precipitation driving the Elbe floods earlier this millennium as 'high-intensity' or 'large amounts'? (actually, I think it was a combination of both....). I would avoid suggesting that more intense summer showers will cause the Rhine to flood Cologne.

Discussion point to consider: Your 30-year time slices from which you determined the flood maxima may not include the very extremes that are relevant for flood protection - in the box plot you see a few isolated extremes that occurred in your simulations. To what extent do you think this affects your overall message?

P15, L10. Under recommendations: Would calibration on observed extent of snow cover using RS support calibration of snow melt modules to support these analyses? To what extent would precipitation falling on a snow cover further enhance melting of a snow cover (so: not snow melt not only depends on T, but also on warm precipitation water) - and would we need to consider that in modeling? Lakes: of course these buffer flow in dry periods, but that was not the focus of your paper, it might be interesting to see their role in generating peak flows.

For policy makers / river managers it may be relevant to see a conclusion on whether we should anticipate changes in summer floods - as the use of the river banks (agriculture, tourism) has a strong seasonality.

---

## Author Comment (AC1) · 11 Jan 2021

**hess-2020-605**
**Responses to anonymous referee 1**

Erwin Rottler, Axel Bronstert, Gerd Bürger and Oldrich Rakovec

January 11, 2021

Dear Anonymous Reviewer 1,

thank you very much for reviewing our manuscript. We are very grateful for your comments and suggestions. In the following, we provide detailed responses to all your comments.

On behalf of all authors,

Sincerely,

Erwin Rottler

**Contents**

**1 General Comment**

Scientific terms and abbreviations should be the same in figures, tables and the text. E.g. in Fig 5 the term "discharge" is used but in the text "streamflow". in Fig 6 and its captions is written "melt magnitude" but in table 2 and the text the abbreviation Smax14. In table 2 Smax14 is singular, in the text sometimes plural. In the figures, tables and their captions should be checked if the same terms are used as in the text. I suggest to give the explanation of abbreviations in the method chapter, in table 2 and in the captions of the figures (but there only in brackets).

Thank you for pointing at this issue. We will harmonise the expressions including figures and tables thoughout our manuscript.

**2 Specific comments**

**2.1 Comment 1 - Data and Methods**

The expression "sub-basin upstream gauge Basel" is quite complicate and, as I think, not necessary. I suggest to explain, that some of the investigated variables refer to the gauge and others to the basin. And then I would refer only to "gauge Basel" or "basin of gauge Basel"). Same for Cochem. Especially in the later chapters the long term of "sub-basin upstream gauge Basel" is a bit confusing.

Yes, we also discussed this internally already. We agree that the expression "sub-basin upstream gauge [...]" is a bit bulky/cumbersome. We will make it shorter and concise in the revised version.

**2.2 Comment 2 - Data and Methods**

P7, Fig3: Missspelling in the last point (elevation and solid precipitation)

Yes. You are right. We will correct this typing error.

**2.3 Comment 3 - Data and Methods**

P8, Table 2: the two last variables are not listed on P6, L11-L13. For me it is not clear what is meant with "melt elevation" especially when it comes to the units (see also Fig 6). For more clearity one could give the units in the table that also helps to understand if it is a value at a gauge or for a basin.

Thank you for your comment. We will introduce all different variables by extending Tab. 2 with information on their units and scale (gauge or basin).

**2.4 Comment 4 - Results**

It would be helpful, if the figures would be described more systematically. Sometimes exact values are given, sometimes not. Sometimes the results for Basel are described first, sometimes those for Cochem. It would be less confusing for the reader, wenn the order would be always like in the figures.

Yes, a recurring structure closer to the order within the figures can facilitate the reading of the manuscript. We will re-structure our text accordingly.

**2.5 Comment 5 - Results**

P8, L1: In my opinion the first two sentences contain already important results and therefore should be more precise and with more information. E.g. like this: "According to the model simulations the changes are largest at gauge Basel (Fig. 5a). Here, the median of discharge magnitude increases from 2500(?) m 3 /s in the historic period to 2700 (?) m 3 /s supposing a warming of 1.5 degree. Furthermore, the highest floods are higher than in the historic period. However, this increase in discharge is not linear with temperature rise....".

Thank you for your observation. Yes, the first two sentences on page 8 are short, but contain important information already. We will rephrase them following you recommendations.

**2.6 Comment 6 - Results**

P8, L4: The results for Cochem and Cologne (Fig 5b and 5c) should be briefly mentioned already here (e.g. Cochem shows only very little increase of maximum discharge with increasing temperature).

We will include information on gauges Cochem and Cologne here.

**2.7 Comment 7 - Results**

P8, L5: The figure reference is not correct. Should be Fig. 5d to 5f.

We will add the correct reference here.

**2.8 Comment 8 - Results**

P8, L8: I see a signal of change for Cochem: annual maxima seems to be a bit earlier (probably due to less solid fraction in the Vosges as prooved in fig 6) .

Yes, there seems to be the tendency that with rising temperatures annual streamflow maxima observed at gauge Cochem more and more are restricted to winter (December to February). With rising temperatures fewer peaks seem to occur in spring. We will take a closer look into this during the revision.

**2.9 Comment 9 - Results**

P8, L13: Wrong word? "...runoff contribution of snowmelt of more the 20%..." should probably be "...runoff contribution of snowmelt of more than 20%...".

Reviewer is right, it should be "more than". We will correct this.

**2.10 Comment 10 - Results**

P8, L15: Here, Smax14 is plural and in the sentence before singular.

In the sentence before, we refer to the "median of Smax14" and later on we talk about all

Smax14 values. We will scan through our manuscript to make sure that there are no mix ups between singular and plural with regard to the abbreviations.

**2.11 Comment 11 - Results**

P8, L16: "solid Pmax5" is probably wrong, "total Pmax5" is probably meant as shown in the Figure and als logically (solid Pmax5 decreases with temperature rise).

Thank you for pointing at this. Yes, it should be "total Pmax5". We will correct this.

**2.12 Comment 12 - Results**

P8, L22: Value of solid fraction should also be given for the historic period (80%), so that the reader do not have to look for it in the figure.

We will include these values to the corresponding paragraph.

**2.13 Comment 13 - Results**

P11, L1: In think the information on highest ETmax14 is not relevant for the purpose of this study that is flood seasonality. This sentence should be deleted.

Yes, the exact number of the ETmax14 seems unnecessary here. In our attempt to establish recurring structures in the individual section, well will also re-phrase this paragraph.

**2.14 Comment 14 - Discussion and Conclusion**

P12, L3: Delete sixth word "in".

Thank you, we agree one word definitely is too much. We will correct the sentence appropriately.

**2.15 Comment 15 - Discussion and Conclusion**

> P14, L32: Concerning precipitation intensity (see also Abstract): I think the study does not show, that the precipitation intensity increases. The rainfall intensity (or I would say "rainfall amount" because I associate intensity with shorter event of up to 72 h) increases due to higher fraction of liquid precipitation. Less snow, more rain... But is the total amount of precipitation increasing? If yes, I missed this result before.

Yes, you are right. It is better to talk about "rainfall amounts" or "totals", not "rainfall intensities". With regard to rainfall, we currently investigate the five days before an event. Our results indicate, that the 5-day sums are changing. We will change our manuscript accordingly.

**2.16 Comment 16 - Discussion and Conclusion**

> P15, L14: Here one could mention the lake of Constance, that has a considerable influence on the flood magnitude at gauge Basel. The lake of Constance is a big storage for the snow melt from the "Alpenrhein". By the way, the lake is not shown on the maps (and I wonder if it is part of the mHM). At least one should briefly mention the lake and its effect in general.

We will include further information on lakes and their influence on flooding in the Rhine Basin. Yes, the Lake Constance represents a large storage and has a strong dampening effect. We will extend corresponding paragraphs in the discussion and conclusion accordingly.

---

## Author Comment (AC2) · 11 Jan 2021

**hess-2020-605**
**Responses to anonymous referee 2**

Erwin Rottler, Axel Bronstert, Gerd Bürger and Oldrich Rakovec

January 11, 2021

Dear Anonymous Reviewer 2,

thank you very much for reviewing our manuscript. We are very grateful for your comments and suggestions. In the following, we provide detailed responses to all your comments.

On behalf of all authors,

Sincerely,

Erwin Rottler

**Contents**

**1 General Comment**

Overall quality of the preprint: A well-structured paper that supports earlier results and adds additional insights into the shifts of flood genesis under climate change. The latter could be highlighted a bit more in the abstract and other parts of the text (suggestions under "specific comments). Principle review criteria (scientific significance, scientific quality, and presentation quality) are generally evaluated as "good". Suggest to accept with revisions.

We will revise our abstract and corresponding paragraphs in the discussion and conclusion and focus more on the additional insights rather than already known results.

**2 Specific comments**

**2.1 Comment 1**

Page 1, Abstract - change request: The abstract describes basic mechanisms of the flow regime of the Rhine River in a warmer climate. This is neither new – cf. e.g. to Kwadijk Romans (1995; https://link.springer.com/article/10.1007/BF01093854) - nor the core of the study presented here. It is suggested (a) to highlight a set of change signals of the hydrological characteristics you evaluated (number of years with snowmelt fraction above a threshold or lift of "melt elevation" etc.) and/or (b) focus more on the hypothesized new flood type superimposing rainfall- und snowmelt-induced runoff.

Thank you for pointing at this. We will revise the abstract and focus more on the specific changes in hydrological characteristics we detect.

**2.2 Comment 2**

Page 1, line 16 - suggestion: The term "current climate crisis" has a political flavor.

Indeed. We will think about a better term and get rid of any political flavor.

**2.3 Comment 3**

Page 2, lines 1ff - suggestion: Add reference to IPCC SROCC

Yes, we will include this reference.

**2.4 Comment 4**

Page 3, lines 9ff - change request: Please add here, that hydrological processes are modelled at 5 km grid resolution (referring to page 5). Otherwise the reader waits for a final downscaling step of met. data to the 500 m grid of mHM.

We will add this information.

**2.5 Comment 5**

Page 3, line 16 – change request: The quoting used here reads like the authors do not understand what this part of the procedure/sentence means. Is that the intention here? The bias correction procedure is important when dealing with peak flow analyses (and heavy precipitation). Please rephrase.

We agree. The bias correction is a very important step. We will rephrase this paragraph and include additional information on the downscaling and bias correction of the climate model data.

**2.6 Comment 6**

Page 3, line 22ff – change request: Please explain how you treated the catchment upstream of Basel. As it reads now you would end up with two parameter sets; one from the calibration of Basel, one from the calibration of Lobith (also containing the catchment upstream of Basel). Please clarify, which parameter set you used for the overlapping part of the catchment or if you used individual model set ups for each gauging station.

During the multi-basin calibration, we attain one parameter set, which we use for the entire basin. We use one model set up. We will rephrase this paragraph to make it more clear.

**2.7 Comment 7**

> Page 6, line 17f – change request: Please add some details on your experience concerning the 14-day time window for snowmelt and evaporation. Is it based on investigations of historical floods? Or on model simulations?

You are right. We need to explain this better. We will extend this section and add more information on the selection of the window width we used.

**2.8 Comment 8**

> Page 6, line 25f – change request: The flow regime at gauge Cologne is usually regarded as "complex" regime containing "nival" and "pluvial" characteristics. This should be added here. Now, the gauge is described as another pluvial example.

We will add this information here.

**2.9 Comment 9**

> Page 7, Figure 3 – change request: The map shows the Rhine River basin up to Lobith, not the entire Basin. This should be added in the scheme and/or caption.

You are right. In Fig. 2 we still explicitly mention it, but in the scheme in Fig. 3 this information is missing. We will update manuscript accordingly, to eliminate this confusion.

**2.10 Comment 10**

> Page 7, Figure 4 – suggestion: For reasons of consistency it is suggested not to introduce an additional reference period here (1971-2016). The period 1971-2000 should be chosen here as well.

In order to illustrate the long-term runoff seasonality of the gauges observed, we decided to use the 100-year time window 1917-2016. Yes, the usage of the period 1971-2000 would make it more consistent with the other analyses so we will change this during the revision.

**2.11 Comment 11**

> Page 8, line 11ff – suggestion: For some readers it may be interesting to note that according to your results there will still be some snowmelt at gauge Cochem even in a 3 ◦ C warmer world. Suggest to add this point.

We will add this specific information into our revised manuscript.

**2.12 Comment 12**

> Page 8, line 13 – suggestion: The units of the variables could be changed to give a better "grip" of the results. For example, "the number of streamflow maxima having an estimated runoff contribution of snowmelt of more than 20

We will improve this formulation and rephrase the sentence accordingly.

**2.13 Comment 13**

> Page 8, line 21 – change request: "Decreases in solid precipitation are most prominent in winter" ← That's not surprising because according to your results the historical period shows is no solid precipitation in summer. Rephrase, e.g. referring to meteorological seasons (DJF, MAM, JJA, SON).

We will rephrase this section of the text as suggested.

**2.14 Comment 14**

> Page 9, lines 3 – suggestion: Suggest to repeat here that the timing of the highest annual flow remains unchanged.

We will do so.

**2.15 Comment 15**

Page 10, lines 6f – suggestion: The role of evaporation simulated under climate change conditions strongly depends on the evaporation approach used and the area of interest. Suggest to transport this uncertainty of hydrological modelling by formulating more carefully. For example: "With the approach used here, evaporation seems to play a minor role . . .".

Thank you for pointing at this. Yes, the evapotranspiration approach used is a crucial point. We will rephrase this part.

**2.16 Comment 16**

Page 11, line 6f. – suggestion: It would be also interesting to state already here that snowmelt-driven flooding is possible despite of rising temperatures. At least in low warming levels there may still be relevant snowmelt events. This follows only two pages later.

Yes. That is a good idea. We will add this information here.

**2.17 Comment 17**

Page 11, line 11 – change request: The hypothesis mentioned in the introduction was on flood risks resulting from the overlap of nival and pluvial peak flows. Here, the focus is on snow-melt driven floods only. Check consistency.

The idea of earlier snowmelt-driven floods is part of the hypothesis of a possible overlap of nival and pluvial peaks. We will rephrase this sentence to make it more clear.

**2.18 Comment 18**

Page 12, figure 8 – suggestion: Add the range that is displayed by the boxes.

We will improve this figure and its display ranges.

**2.19 Comment 19**

> Page 12, line 1f – suggestion: Suggest to stay focused on floods and skip low flows.

Yes, we should focus of floods and be careful talking too much about low flows.

**2.20 Comment 20**

> Page 12, line 3ff – change request: In this paragraph it is advisable to be very clear about (a) the statistics (e.g. to avoid confusion between monthly and annual stream flow maxima) and (b) the gauge/regime that is discussed. Otherwise the reader will be lost. For example the statement that "with rising temperatures, most flood events will occur in winter" does obviously not relate to Basel/nival regimes. This has to be more transparent.

Thank you for pointing this out. We need to improve this paragraph. We will rephrase this paragraph and also look at other parts of the result and discussion section.

**2.21 Comment 21**

> Page 14, line 7 – change request: In how far are peak elevations (here: 1300 m a.s.l.) and the related processes interpreted here reflected in the hydrological model, given the 5 km grid resolution? Please add this to the method description (page 3f.).

We will include further information on this in the method section.

**2.22 Comment 22**

> Page 14, line 30ff. – suggestion: cf. comment on abstract. Suggest to refocus this paragraph in the same way.

Following your suggestions on how to improve the abstract, we will also rewrite this paragraph.

**2.23 Comment 23**

> Page 15, line 14f. – suggestion: Suggest to stay focused on floods and skip low flows. Lakes and reservoirs play an important role for high flow, too. If they are not yet implemented in the model, this should be mentioned in the methods chapter (page 3f.)

We will include this information into the method section.

**3 Technical correction**

**3.1 Comment 24**

> General comment: It was difficult to print the pdf. Presumably one of the graphs is oversampled – please check.

Yes. Thank you for hinting at this. The problem is Fig. 7 itself. We will export the figure in a different format to ensure its accessibility.

**3.2 Comment 25**

> Page 8, line 5 – change request: Figure 5b contains no information on the timing of runoff maxima. Wrong reference. Please correct (-> 5d?) and repeat the reference to "Basel" in the text/line 5.

Yes, the reference is not correct. We will change this.

**3.3 Comment 26**

> Page 8, line 13 – change request: "more the" -> "more than"

Agree. Needs to be "more than".

**3.4 Comment 27**

Page 9, Figure 5 – change request: The horizontal grid lines do only occasionally match the tick marks. Please correct.

We will correct this in the figure.

**3.5 Comment 28**

Page 10, Figure 6 – change request: The horizontal grid lines do only occasionally match the tick marks. Please correct.

We will correct this in the figure.

**3.6 Comment 29**

Page 11, line 6 – change request: Replace "Smax14" by plain text.

Yes, using plain text here is better. We will rephrase the sentence.

**3.7 Comment 30**

Page 12, Figure 8 – change request: The horizontal grid lines do only occasionally match the tick marks. Please correct.

We will correct this in the figure.

**3.8 Comment 31**

Page 13, Figure 9 – change request: The horizontal grid lines do only occasionally match the tick marks. Please correct.

We will correct this in the figure.

**3.9 Comment 32**

Page 24, Figure B1 – change request: The horizontal grid lines do only occasionally match the tick marks. Please correct.

We will correct this in the figure.

**3.10 Comment 32**

Page 25, Figure C1 – change request: The horizontal grid lines do only occasionally match the tick marks. Please correct.

We will correct this in the figure.

---

## Author Comment (AC3) · 11 Jan 2021

**hess-2020-605**
**Responses to anonymous referee 3**

**Erwin Rottler, Axel Bronstert, Gerd Bürger and Oldrich Rakovec**

January 11, 2021

Dear Anonymous Reviewer 3,

thank you very much for reviewing our manuscript. We are very grateful for your comments and suggestions. In the following, we provide detailed responses to all your comments.

On behalf of all authors,

Sincerely,

Erwin Rottler

**Contents**

**1 General Comment**

This paper analyses future changes in flood seasonality in the Rhine River Basin at three different global warming levels using the mesoscale Hydrological Model (mHM). The paper is well structured and written, considers earlier work quite well, and provides new insights in flood seasonality changes under climate change for the Rhine basin. Finally, the authors list some next steps to improve the modelling approach as including a glacier module or reservoir and lake functionality.

Thank you for reviewing our manuscript. In the following, detailed responses to all your comments.

**2 Specific comments**

**2.1 Comment 1**

Data and Methods: suggest to include that the model does not include a glacier and lakes module. For the basin upstream Basel not including lakes can have quite some effect. Now, this becomes only clear at the end of the Conclusions section.

We will include this information earlier, i.e., in the method section.

**2.2 Comment 2**

Page 3, line 15, please describe the downscale and bias correction in more detail. The sentence "adjusts the monthly mean and daily variability of simulated climate data to observations." does not describe how this was done.

Thank you for pointing at this. We will extend this paragraph and provide additional information on the downscaling and bias correction.

**2.3  Comment 3**

> Page 3, lines 22-25, The calibration procedure could be described in more detail. 1) For example why was the gauge Lobith also included in the calibration procedure? With MPR, one could have choosen for example three smaller sub-basins to find how well parameters are transferable to the larger basin scale. This makes the calibration more efficient, and would also provide an interesting result (although I understand this is not the focus of the paper, it is an important aspect of this study). 2) What were the specific DDS settings (e.g. number of function evaluations)? Please add these to the text. 3) Finally, how many model parameters were calibrated? At least this gives the reader some insight into the model complexity.

We will provide more information on our multi-basin calibration approach and model parameters calibrated.

**2.4  Comment 4**

> Page 14, line 5. Suggest to change "increased precipitation intensity" to amount, the analysis is about a monthly time scale, so probably better to use amount and not intensity.

Yes, you are right. "Precipitation intensity" can be a bit misleading in our case. We will think of a better term focusing on the "amount" / "totals".

**2.5  Comment 5**

> Page 15, line 10 and lines 13-14: Please add this reference as an example: https://agupubs.onlinelibrary.wiley.com/doi/abs/10.1029/2019WR026807, for a mod- elling approach also applied to the Rhine basin that already includes a glacier and lake module.

We thank the Reviewer for this very interesting article! We are happy to include it into discussion of our manuscript.

**3 Technical corrections**

**3.1 Comment 6**

Fig 7. change "elvation" to "elevation"

Thank you, we will correct this typo.

**3.2 Comment 7**

Page 8, Table 2, change "ration" to "ratio"

Thank you, we will correct this typo.

---

## Author Comment (AC4) · 11 Jan 2021

**hess-2020-605**
**Responses to anonymous referee 4**

Erwin Rottler, Axel Bronstert, Gerd Bürger and Oldrich Rakovec

January 11, 2021

Dear Anonymous Reviewer 4,

thank you very much for reviewing our manuscript. We are very grateful for your comments and suggestions. In the following, we provide detailed responses to all your comments.

On behalf of all authors,

Sincerely,

Erwin Rottler

**Contents**

**1 General Comment**

> The paper addresses a highly relevant subject on changes in flood seasonality in the Rhine basin, by analysing how climate change affects different components of the water balance and their aggregated effect on peak flows. The authors elegantly demonstrate by means of model simulations for different GCMs and climate scenarios the contributions of snow melt and rainfall-driven runoff to peak flow generation over the seasons. These analyses form a relevant contribution to earlier studies on the impacts of climate change on peak flows in the Rhine basin, provide nice insight in the underlying contributions of rainfall and snowmelt, and indicate their effects on time shifts in peak flow occurrences. The paper's title well covers the contents; the paper is well-structured, clearly presents its results in text and figures, and the interpretations, discussion and conclusions are well supported by the results. I would rate the significance and quality of the paper 'Good'.

Thank you for reviewing our manuscript. In the following, detailed responses to all your comments.

**2 Specific comments**

**2.1 Comment 1**

> Fig 1: I would suppose that the nival peak not only shifts to earlier in the season but also becomes smaller under CC - as there will be less total snow accumulation over the winter season.

You are right. We need to re-think the figure and improve the description of the hypothesis. We will improve the scheme and corresponding paragraphs in the text.

**2.2 Comment 2**

> Line 30: 1.5, 2 and 3 degrees warming - relative to 1970 - 2000

We investigate 1.5, 2.0 and 3.0 °C global warming levels relative to pre-industrial levels. The period 1971–2000 is assumed to be warmer by 0.46 °C compared to pre-industrial levels already. We will add another sentence on this into the method section to avoid any misunderstandings.

**2.3 Comment 3**

> Section 2: provide a bit more information on: which parameters did you calibrate? In particular you have a detailed representation of crops and soil types, did you use reference values in all cases, or did you do any calibration here? How did you choose LAI values for different vegetation types and seasons and latitude? How did you perform the bias correction to GCMs for future climates?

Thank you pointing this issue. We will include additional information on our multi-basin calibration, soil types, LAI values and the bias correction of the GCM data.

**2.4 Comment 4**

> P4, L11: assess -> assesses

Thank you, we will correct this typo.

**2.5 Comment 5**

> P4, L13: bases -> is based

Thank you, we will correct this typo.

**2.6 Comment 6**

> P6, L6-9. Do I understand here that the projection times of the periods where the 'targeted' warming was reached was different for each realisation? And with different RCPs you may reach the same warming at different moments (e.g. 1.5 degree under RCP 8.0 early in the century, and RCP2.6 only late) - but to what extent are these scenarios different in your simulations (associated P?). Can you indicate which are the according time horizons used in your simulations?

Yes, you understand correctly. Different GCM-RCP combinations reach the same warming at different moments. "A detailed description of the determination of warming levels is given in the supplementary material of Thober et al. (2018)" (Page 6, Line 9). We will extend this paragraph and add more information on the determination of warming levels.

**2.7 Comment 7**

P6, L14: for the Rhine basin as a whole (e.g. Cologne) a 5 day-period for the precipitation sum seems quite short to generate extreme floods, in particular in view of saturating the soil and travel time of peaks from tributaries.

We will take a closer look into this and better explain the selection of the window widths for precipitation, snowmelt and precipitation.

**2.8 Comment 8**

P6, L19: river discharge at Basel is considerably dampened by the effects of the Swiss lakes. For that reason, earlier studies focused on catchments upstream of the lakes (e.g. Murg, Thur). Can you indicate to what extent timing and maxima of small peaks - in particular after a dry period with low lake levels - are affected by this?

Yes, discharge upstream gauge Basel is considerably influenced by the large Swiss lakes. We will extend our discussion section and add more information on the dampening effect of large lakes.

**2.9 Comment 9**

Figure 5: Please indicate on how many runs each histogram is based. From the methods I read how many GCMs reached each warming, with only 8 of them reaching 3 degrees, but this is not clear for the other histograms. To what extent do different numbers of realisation result in different occurrences of highest extremes, and did different GCMs result in different extremes under - in spite of bias correction?

We will try to find a way to include the number of GCM-RCP combinations reaching each warming level directly into the figure(s). Yes, you are right. Caution has to be exercised, as the different warming levels base on different numbers of realisations. We display both boxplots and histograms (probability densities) to get a comprehensive insights and try to be cautions when interpreting results. So far, we did not encounter any abnormal differences among GCMs and extremes simulated.

**2.10 Comment 10**

> Fig 5; P8, L9: whereas both for Basel and Cochem there is a decline in the timing of summer maxima for higher temperatures, Cologne shows a small peak emerging around DOY 250 - can you explain this? Consider using the same horizontal scale for figs 5d-f.

Yes, there seems to be a small peak emerging around DOY 250 in the histograms of gauge Cologne. However, when considering such small changes at gauge Cologne, also the tendency toward more peaks in summer at gauge Cochem might need to be considered. We will take a look into this. Changes are only small, but might still be worth discussing. Yes, a common horizontal scale is a good idea.

**2.11 Comment 11**

> P10, L4: is detected -> are detected

Thank you, we will correct this typo.

**2.12 Comment 12**

> P10, fig 6 (k,l): by displaying annual maxima distributions we indeed can see how these shift over time, but we cannot see how shifts evapotranspiration maxima link to peak flows, as the connection to the flood events is lost: we cannot see how much was the 'reduction' of the annual peak flow maxima due to evapotranspiration loss (as you do in fig 6 ab indicating the 'contribution' of snowmelt to the annual maxima). It makes sense that under a warming climate annual maximum evapotranspiration goes up - but if that happens in summer when floods never arise it is hard to judge the role of evapotranspiration in changing peak flows. In fig 9g we can see that the contribution of evapotranspiration change is small indeed.

We agree. The current analysis does not allow to directly link changes in evapotranspiration to peak flows. Yes, the investigation of monthly maxima can give a first insight, however, does not fully address this issue. We will think about this and improve our manuscript with regard to this aspect.

**2.13 Comment 13**

> P11: Discussions -> discussion

Thank you, we will correct this typo.

**2.14 Comment 14**

> P11, L4: ....diminish seasonal snow covers -> please add in a few words the key aspects of that: the total volume, the duration and the timing of melt.

Thank you for this hint. We will include additional information here to improve clarity.

**2.15 Comment 15**

> P11, L6: Smax14 is singular.

We refer to snowmelt sums displayed in the indivudial boxes of the months. We will rephrase this part.

**2.16 Comment 16**

> P11, L8: 'forward' -> in first time use, explicitly explain that you mean: 'earlier in the year'

We will rephrase this accordingly.

**2.17 Comment 17**

> P11, L10, 13-14: two factors may play a role: the timing of melting, and the amount of snow that has accumulated so far to be available for melting - you do not indicate the maximum amount of snow that has accumulated by the end of the season to be available for melt.

We will rephrase this sentence to avoid any misunderstandings. Yes, reduced snow accumulation also plays an important role.

**2.18 Comment 18**

> P11, L15. I do not see a contradiction suggested by using 'however'. Actually, you change subject here to low flow situations, as caused by disappearing glaciers and intensified evapotranspiration, which becomes different from 'lower maxima',

We agree with Reviewer that using "However" is not correct. We will rephrase this sentence.

**2.19 Comment 19**

> P12. l6 (and in the rest of the paper, in particular in the conclusion, P14, L32): I am not sure whether you should formulate this as 'intense rainfall events' in mm per hour and use this formulation for both summer and winter. 'High intensity' rather relates to high intensity summer storms - as you indicate in lines 12-15 here, but for the winter season I would not formulate that as intensity. Moreover, you consider in your study accumulated precipitation sums over 5 days - that is an amount, not an intensity.

Yes, you are right. We will change this and not call it "rainfall intensity". We will find a different term that describes the variable investigated better, e.g. "rainfall totals" or "rainfall amounts".

**2.20 Comment 20**

> P12, L12. Here you make a relevant statement - that the summer extremes were still supported by snow melt from the Alps to produce their maxima - I presume that that has been derived from the associated historic descriptions. With reduced snow melt in late spring this would indeed reduce the risk of summer floods. Conversely, higher temperatures under future climate change may lead to more intense summer precipitation - still causing higher peak flows. Here we may encounter the questions: does the latter only hold for smaller catchments within the basin, or do these still feed large floods to Cologne? And some GCMs show a N-S difference in precipitation change (some even the signal) across the Rhine basin: is that the case in your experiments? From the histograms in fig 6g this is hard to derive. It would affect probabilities of extreme summer floods, as we experienced in early 2000s in the Elbe.

Yes, this is a very important point. For large basins, such as gauge at Cologne, local convective storms hardly play any role. Our results indicate an increase in 5-day rainfall amounts in the investigated sub-basins, also for summer. We will take a closer look into this and extend our manuscript accordingly.

**2.21 Comment 21**

P14, L3: It is not a true 'interaction' between snow fall and precipitation, but their effects are counterbalancing - as you explain in the following lines.

We will rephrase this and avoid the term "interaction" here.

**2.22 Comment 22**

P14, L9: hint -> suggest

Thank you, we will modify this.

**2.23 Comment 23**

P14, L14: originate - > originates; During this period, we have experienced already over 0.5 degree of warming, so it would be interesting to know what the average for the past few decades would be.

Yes, the numbers we present from Stahl et al. (2016) refer to the historic period 1901–2006. Stahl et al. (2006) provide both long-term averages and figures with the time series of the runoff components: Fig. 4 in
https://chr-khr.org/en/file/1057/download?token=Zg6SY04i
Yes, due to rising temperatures within the 20th century, there possibly has been changes in the fraction of snowmelt contributions to runoff already. In our opinion, Fig. 4 also highlights the strong annual to decal variability of the relative fraction of the streamflow components. We will rephrase this part accordingly.

**2.24 Comment 24**

P14, L33: 'intense precipitation events' (see earlier comment): would you describe the precipitation driving the Elbe floods earlier this millennium as 'high-intensity' or 'large amounts'? (actually, I think it was a combination of both....). I would avoid suggesting that more intense summer showers will cause the Rhine to flood Cologne.

Indeed. To avoid any misunderstandings, we will replace the term "intensity". We will scan through our manuscript and rephrase all corresponding sentences.

**2.25 Comment 25**

> Discussion point to consider: Your 30-year time slices from which you determined the flood maxima may not include the very extremes that are relevant for flood protection - in the box plot you see a few isolated extremes that occurred in your simulations. To what extent do you think this affects your overall message?

Yes, with regard to flood protection, the very high values are of great interest. In this analysis, we focused on the hypothesis of a merging of the flood regimes that might result in the creation of such extreme events. We are mostly interested in temporal shift and changes in magnitudes. In this regard, we focus on changes in underlying flood-generating processes. "Isolated extremes" do not influence our results and overall message. However, we will include this point of singly very strong extremes and their significance into our discussion.

**2.26 Comment 26**

> P15, L10. Under recommendations: Would calibration on observed extent of snow cover using RS support calibration of snow melt modules to support these analyses? To what extent would precipitation falling on a snow cover further enhance melting of a snow cover (so: not snow melt not only depends on T, but also on warm precipitation water) - and would we need to consider that in modeling? Lakes: of course these buffer flow in dry periods, but that was not the focus of your paper, it might be interesting to see their role in generating peak flows.

Yes, using a multi-variable calibration including satellite-based snow cover maps provides a very good opportunity to further improve the simulations. We will include this information into the conclusion. The snow module currently available in mHM is based on a degree-day approach. As we describe in the method section: "In order to account for snowmelt following the energy input from liquid rainfall, degree-day factors are increased depending on the amount of liquid precipitation. Degree-day factors only can increase to a certain threshold value." Hence, mHM already has a (simple) way of addressing the energy input through liquid rain. The implementation of a physically-based snow routine might improve this aspect and addresses rain-on-snow events better.

**2.27 Comment 27**

For policy makers / river managers it may be relevant to see a conclusion on whether we should anticipate changes in summer floods - as the use of the river banks (agriculture, tourism) has a strong seasonality.
* * *
Yes. We will try to extend our discussion/conclusion part with regard to this aspect.

---

## Referee Comment (RC5) · Anonymous Referee #4 · 12 Jan 2021

Dear authors, Thanks for your replies to my review comments, and I agree with your responses; I look forward to seeing the revised version - it is a really nice paper.

---

## Referee Comment (RC6) · Anonymous Referee #2 · 12 Jan 2021

Dear Authors. I see all points covered. Thank you, and "Thumbs up".

---

## Author Comment (AC5) · 19 Jan 2021

Dear Anonymous Referee #2,

thank you very much for your comments. We hope to get the chance to revise and improve our manuscript following your suggestions.

On behalf of all authors,

Sincerely,

Erwin Rottler

---

## Author Comment (AC6) · 19 Jan 2021

Dear Anonymous Referee #4,

thank you very much for your comments and suggestions. We hope to get the chance to revise and improve our manuscript.

On behalf of all authors,

Sincerely,

Erwin Rottler
* * *

---

## Author Response (AR1)

**hess-2020-605**
**Revision**

Erwin Rottler, Axel Bronstert, Gerd Bürger and Oldrich Rakovec

January 29, 2021

Dear Editor,

we are very happy to get the chance to revise our manuscript. In the following, we list all comments of the four reviewers, our responses and how we changed our manuscript accordingly. When specifying page/line numbers, we refer to the 'Marked-up manuscript version' below, where all additions and deletions in comparison to the previous version are marked. Should you have any further questions or comments, please do not hesitate to contact us. I would like to thank you for considering our work.

On behalf of all authors,

Sincerely,

Erwin Rottler

**Contents**

**1 Anonymous referee 1**

**1.1 General Comment**

> Scientific terms and abbreviations should be the same in figures, tables and the text. E.g. in Fig 5 the term "discharge" is used but in the text "streamflow". in Fig 6 and its captions is written "melt magnitude" but in table 2 and the text the abbreviation Smax14. In table 2 Smax14 is singular, in the text sometimes plural. In the figures, tables and their captions should be checked if the same terms are used as in the text. I suggest to give the explanation of abbreviations in the method chapter, in table 2 and in the captions of the figures (but there only in brackets).

We updated all figures and Table 2 and changed the manuscript accordingly. Now we explain the variables investigated in the method section, Table 2 and throughout the figure captions.

**1.2 Specific comments**

**1.2.1 Comment 1 - Data and Methods**

> The expression "sub-basin upstream gauge Basel" is quite complicate and, as I think, not necessary. I suggest to explain, that some of the investigated variables refer to the gauge and others to the basin. And then I would refer only to "gauge Basel" or "basin of gauge Basel"). Same for Cochem. Especially in the later chapters the long term of "sub-basin upstream gauge Basel" is a bit confusing.

We scanned through our manuscript and now avoid using this bulky expression. Yes, most of the time writing "gauge Basel" is sufficient and does not slow down the reading flow.

**1.2.2 Comment 2 - Data and Methods**

> P7, Fig3: Missspelling in the last point (elevation and solid precipitation)

We corrected this typo. Now it is "elevation".

**1.2.3 Comment 3 - Data and Methods**

> P8, Table 2: the two last variables are not listed on P6, L11-L13. For me it is not clear what is meant with "melt elevation" especially when it comes to the units (see also Fig 6). For more clearity one could give the units in the table that also helps to understand if it is a value at a gauge or for a basin.

We updated Table 2. The table only includes "variables investigated on sub-basin level" (see table caption). Furthermore, we harmonised the table, figures and text with regard to abbreviations used and give an explanation of the variables investigated in the table, figure captions and method section.

**1.2.4 Comment 4 - Results**

> It would be helpful, if the figures would be described more systematically. Sometimes exact values are given, sometimes not. Sometimes the results for Basel are described first, sometimes those for Cochem. It would be less confusing for the reader, wenn the order would be always like in the figures.

We have restructured our result section and we believe that now we describe our results more systematically. With regard to streamflow, we always start with gauge Basel and work our story line via Cochem to Cologne. The presentation of all other results follows the order of the figure panels. In addition, we have further included subsection headings to facilitate the navigation in the result section.

**1.2.5 Comment 5 - Results**

> P8, L1: In my opinion the first two sentences contain already important results and therefore should be more precise and with more information. E.g. like this: "According to the model simulations the changes are largest at gauge Basel (Fig. 5a). Here, the median of discharge magnitude increases from 2500(?) m 3 /s in the historic period to 2700 (?) m 3 /s supposing a warming of 1.5 degree. Furthermore, the highest floods are higher than in the historic period. However, this increase in discharge is not linear with temperature rise....".

We revised this part of the result section and included additional information (P9 L13-17).

**1.2.6 Comment 6 - Results**

P8, L4: The results for Cochem and Cologne (Fig 5b and 5c) should be briefly mentioned already here (e.g. Cochem shows only very little increase of maximum discharge with increasing temperature).

Information on changes in annual streamflow maxima is presented in the sentences that follow (P9 L20).

**1.2.7 Comment 7 - Results**

P8, L5: The figure reference is not correct. Should be Fig. 5d to 5f.

We corrected this sentence: "At gauge Basel, annual streamflow maxima occur throughout the year (Fig. 5 d)". (see P9 L18)..

**1.2.8 Comment 8 - Results**

P8, L8: I see a signal of change for Cochem: annual maxima seems to be a bit earlier (probably due to less solid fraction in the Vosges as prooved in fig 6) .

Yes, there maybe is a tendency towards earlier annual streamflow maxima at gauge Cochem (Fig. 5 e). However, taking a close look at the histograms, this signal is not clear. We hesitate to explicitly mention this possible tendency and think that it is better to focus the discussion on the clear signals that we attain. We prefer to be cautious here not to over-interpret our results.

**1.2.9 Comment 9 - Results**

P8, L13: Wrong word? "...runoff contribution of snowmelt of more the 20%..." should probably be "...runoff contribution of snowmelt of more than 20%...".

We corrected this typo.

**1.2.10 Comment 10 - Results**

P8, L15: Here, $S_{max}14$ is plural and in the sentence before singular.

We updated Tab. 2 and screened through the text in order to make sure that we always use the correct formulation.

**1.2.11 Comment 11 - Results**

P8, L16: "solid $P_{max}5$" is probably wrong, "total $P_{max}5$" is probably meant as shown in the Figure and als logically (solid $P_{max}5$ decreases with temperature rise).

We corrected this sentence: "In both sub-basins, liquid and total $P_{max}5$ [...]"

**1.2.12 Comment 12 - Results**

P8, L22: Value of solid fraction should also be given for the historic period (80%), so that the reader do not have to look for it in the figure.

We included values of solid fraction for both gauges and the historic and $3\,^{\circ}$C warming (P11 L6-7).

**1.2.13 Comment 13 - Results**

P11, L1: In think the information on highest $ET_{max}14$ is not relevant for the purpose of this study that is flood seasonality. This sentence should be deleted.

We have included the variable $ET_{loss}$ into our analysis (see Tab.2), to have a direct link between annual streamflow maxima and actual evapotranspiration. In our opinion, information on actual ET helps to understand flood magnitudes/seasonality in the Rhine Basin. Yes, ET does not generate floods, however, ET is a key process in the basin analysed and helps to understand why there are no floods in some seasons. For completeness, we included also monthly $ET_{max}10$ (Fig. 9 g and h).

**1.2.14 Comment 14 - Discussion and Conclusion**

P12, L3: Delete sixth word "in".

We deleted the fifth word "at". Now the sentence reads "Our results indicate that in [...]".

**1.2.15 Comment 15 - Discussion and Conclusion**

P14, L32: Concerning precipitation intensity (see also Abstract): I think the study does not show, that the precipitation intensity increases. The rainfall intensity (or I would say "rainfall amount" because I associate intensity with shorter event of up to 72 h) increases due to higher fraction of liquid precipitation. Less snow, more rain... But is the total amount of precipitation increasing? If yes, I missed this result before.

We removed the expression "intensity" and replaced it with either 'sum', 'amount' or talk about antecedent precipitation. We investigate 5-day precipitation sums (total and liquid). Yes, the amount of precipitation within 5-days is increasing.

**1.2.16 Comment 16 - Discussion and Conclusion**

P15, L14: Here one could mention the lake of Constance, that has a considerable influence on the flood magnitude at gauge Basel. The lake of Constance is a big storage for the snow melt from the "Alpenrhein". By the way, the lake is not shown on the maps (and I wonder if it is part of the mHM). At least one should briefly mention the lake and its effect in general.

We included additional information on the effect of the large lakes in Switzerland and Southern Germany into the discussion (P17 L7-10). The mHM model set-up we use does not include a lake module yet. We specifically mention this in the method section of the revised manuscript (P6 L34).

**2 Anonymous referee 2**

**2.1 General Comment**

> Overall quality of the preprint: A well-structured paper that supports earlier results and adds additional insights into the shifts of flood genesis under climate change. The latter could be highlighted a bit more in the abstract and other parts of the text (suggestions under "specific comments). Principle review criteria (scientific significance, scientific quality, and presentation quality) are generally evaluated as "good". Suggest to accept with revisions.

Thank you very much for your comments and suggestions. In the following, we provide details responses to all your comments.

**2.2 Specific comments**

**2.2.1 Comment 1**

> Page 1, Abstract - change request: The abstract describes basic mechanisms of the flow regime of the Rhine River in a warmer climate. This is neither new – cf. e.g. to Kwadijk Romans (1995; https://link.springer.com/article/10.1007/BF01093854) - nor the core of the study presented here. It is suggested (a) to highlight a set of change signals of the hydrological characteristics you evaluated (number of years with snowmelt fraction above a threshold or lift of "melt elevation" etc.) and/or (b) focus more on the hypothesized new flood type superimposing rainfall- und snowmelt-induced runoff.

We revised the abstract accordingly.

**2.2.2 Comment 2**

> Page 1, line 16 - suggestion: The term "current climate crisis" has a political flavor.

We changed the term and now write "current climatic changes".

**2.2.3 Comment 3**

Page 2, lines 1ff - suggestion: Add reference to IPCC SROCC

We included the IPCC SROCC as reference (P2 L3).

**2.2.4 Comment 4**

Page 3, lines 9ff - change request: Please add here, that hydrological processes are modelled at 5 km grid resolution (referring to page 5). Otherwise the reader waits for a final downscaling step of met. data to the 500 m grid of mHM.

We added the information that "hydrological processes are modelled at 5 km resolution" into this section already (P3 L24).

**2.2.5 Comment 5**

Page 3, line 16 – change request: The quoting used here reads like the authors do not understand what this part of the procedure/sentence means. Is that the intention here? The bias correction procedure is important when dealing with peak flow analyses (and heavy precipitation). Please rephrase.

We included additional information the the GCM data and the ISI-MIP bias correction approach (P4 L3-12).

**2.2.6 Comment 6**

Page 3, line 22ff – change request: Please explain how you treated the catchment upstream of Basel. As it reads now you would end up with two parameter sets; one from the calibration of Basel, one from the calibration of Lobith (also containing the catchment upstream of Basel). Please clarify, which parameter set you used for the overlapping part of the catchment or if you used individual model set ups for each gauging station.

We updated the section on the model calibration and included additional information on the multi-basin approach. During calibration we attain one set of global parameters, which we apply to the entire basin (P5 L4-13).

**2.2.7 Comment 7**

> Page 6, line 17f – change request: Please add some details on your experience concerning the 14-day time window for snowmelt and evaporation. Is it based on investigations of historical floods? Or on model simulations?

We included additional information on the selection of window width in the method section (P8 L5-8). The selection of the 10-day window mainly bases on our previous modelling experience. In our opinion, this window width provide a good estimate for the size of sub-catchments investigated.

**2.2.8 Comment 8**

> Page 6, line 25f – change request: The flow regime at gauge Cologne is usually regarded as "complex" regime containing "nival" and "pluvial" characteristics. This should be added here. Now, the gauge is described as another pluvial example.

Following you recommendation, we added another sentence to this paragraph: "Streamflow at gauge Cologne is characterised by a complex flow regime containing both nival and pluvial characteristics." (P8 L32).

**2.2.9 Comment 9**

> Page 7, Figure 3 – change request: The map shows the Rhine River basin up to Lobith, not the entire Basin. This should be added in the scheme and/or caption.

We added gauge Lobith to the map. To further increase clarity, we extended the figure caption and explicitly mention gauges and sub-basins investigated in detail.

**2.2.10 Comment 10**

> Page 7, Figure 4 – suggestion: For reasons of consistency it is suggested not to introduce an additional reference period here (1971-2016). The period 1971-2000 should be chosen here as well.

We change the time frame investigated in Fig. 4 to 1971–2000.

**2.2.11 Comment 11**

> Page 8, line 11ff – suggestion: For some readers it may be interesting to note that according to your results there will still be some snowmelt at gauge Cochem even in a 3 ∘ C warmer world. Suggest to add this point.

We now provide values for the Sfrac and Smax10 for gauge Cochem (P10 L3-12).

**2.2.12 Comment 12**

> Page 8, line 13 – suggestion: The units of the variables could be changed to give a better "grip" of the results. For example, "the number of streamflow maxima having an estimated runoff contribution of snowmelt of more than 20

We updated this section in the results section. In addition, we also use [%] in the figures (see e.g. Fig. 6).

**2.2.13 Comment 13**

> Page 8, line 21 – change request: "Decreases in solid precipitation are most prominent in winter" ← That's not surprising because according to your results the historical period shows is no solid precipitation in summer. Rephrase, e.g. referring to meteorological seasons (DJF, MAM, JJA, SON).

We rephrased this sentences: "At gauge Basel (Cochem), the solid fraction of precipitation ($P_{solid}$) reaches values of 69.9 % (43.9 %) during winter in the historic time frame (Fig. 7 a and b). Our results indicate that at a 3 °C warming, on average, the fraction of solid precipitation will be reduced to less than 40 % (17 %) at gauge Basel (Cochem) in winter."

**2.2.14 Comment 14**

> Page 9, lines 3 – suggestion: Suggest to repeat here that the timing of the highest annual flow remains unchanged.

To avoid confusion, we decided to not repeat the information of Fig. 5 in this section.

**2.2.15 Comment 15**

> Page 10, lines 6f – suggestion: The role of evaporation simulated under climate change conditions strongly depends on the evaporation approach used and the area of interest. Suggest to transport this uncertainty of hydrological modelling by formulating more carefully. For example: "With the approach used here, evaporation seems to play a minor role . . .".

We rephrased this sentence: "Our model simulation suggest that evapotranspiration only plays a minor role [...]". To improve the understanding of the role of ET, we extended our analysis and directly link actual evapotranspiration and streamflow maxima (Fig. 6 k and l).

**2.2.16 Comment 16**

> Page 11, line 6f. – suggestion: It would be also interesting to state already here that snowmelt-driven flooding is possible despite of rising temperatures. At least in low warming levels there may still be relevant snowmelt events. This follows only two pages later.

Yes, snowmelt will continue to be an important runoff component, particularly in the Alpine areas. Certainly, singular strong snowmelt events always will be possible. We still hesitate to specifically mention this is this part of the discussion. Here, we focus on the the reduction of seasonal snow packs.

**2.2.17 Comment 17**

> Page 11, line 11 – change request: The hypothesis mentioned in the introduction was on flood risks resulting from the overlap of nival and pluvial peak flows. Here, the focus is on snow-melt driven floods only. Check consistency.

We rephrased this sentence: "For the basin until Basel, we can not find indications [...]". Later in the discussion section, we pick up the hypothesis of a potential overlap again and discuss together with changes in precipitation.

**2.2.18 Comment 18**

Page 12, figure 8 – suggestion: Add the range that is displayed by the boxes.

We made sure than horizontal grid lines match the ticks at the y-axis. However, we were not sure how to additionally add 'ranges' displayed by the boxes.

**2.2.19 Comment 19**

Page 12, line 1f – suggestion: Suggest to stay focused on floods and skip low flows.

We still hesitate to remove this sentence from the discussion. Yes, our focus is on floods, however, as detected changes in snow cover and and evapotranspiration also provide information on potential changes in low-flow conditions, we would like to keep this sentence.

**2.2.20 Comment 20**

Page 12, line 3ff – change request: In this paragraph it is advisable to be very clear about (a) the statistics (e.g. to avoid confusion between monthly and annual stream flow maxima) and (b) the gauge/regime that is discussed. Otherwise the reader will be lost. For example the statement that "with rising temperatures, most flood events will occur in winter" does obviously not relate to Basel/nival regimes. This has to be more transparent.

Similar to the order in which we present the streamflow result, we first discuss gauge Basel and move via Cochem to Cologne. With regard to changes in streamflow, we harmonised our manuscript following this order. Furthermore, we made sure to always add a reference to the corresponding result figure. The statement "with rising temperatures, most flood events will occur in winter" does refer to gauge Basel (see Fig. 5 d).

**2.2.21 Comment 21**

Page 14, line 7 – change request: In how far are peak elevations (here: 1300 m a.s.l.) and the related processes interpreted here reflected in the hydrological model, given the 5 km grid resolution? Please add this to the method description (page 3f.).

As we use a 5 km resolution, highest elevations (peaks) and not captured by our model. We

included this information in the method section right after we describe the snow module (P6 L30).

**2.2.22 Comment 22**

> Page 14, line 30ff. – suggestion: cf. comment on abstract. Suggest to refocus this paragraph in the same way.

We updated this part of the conclusion and included additional information.

**2.2.23 Comment 23**

> Page 15, line 14f. – suggestion: Suggest to stay focused on floods and skip low flows. Lakes and reservoirs play an important role for high flow, too. If they are not yet implemented in the model, this should be mentioned in the methods chapter (page 3f.)

We added this information into the method section: "mHM does not include glacier and lake modules yet." (P6 L35).

**2.3 Technical correction**

**2.3.1 Comment 24**

> General comment: It was difficult to print the pdf. Presumably one of the graphs is oversampled – please check.

The problem was Fig. 7 showing the annual cycles. We exported this file in a different format/resolution to avoid the described problems.

**2.3.2 Comment 25**

> Page 8, line 5 – change request: Figure 5b contains no information on the timing of runoff maxima. Wrong reference. Please correct (-> 5d?) and repeat the reference to "Basel" in the text/line 5.

We corrected this sentence: "At gauge Basel, annual streamflow maxima occur throughout the year (Fig. 5 d)". (P9 L18).

**2.3.3 Comment 26**

Page 8, line 13 – change request: "more the" -> "more than"

We corrected this typo.

**2.3.4 Comment 27**

Page 9, Figure 5 – change request: The horizontal grid lines do only occasionally match the tick marks. Please correct.

We updated Figure 5 and made sure that horizontal grid lines match the tick marks.

**2.3.5 Comment 28**

Page 10, Figure 6 – change request: The horizontal grid lines do only occasionally match the tick marks. Please correct.

We updated Figure 6 and made sure that horizontal grid lines match the tick marks.

**2.3.6 Comment 29**

Page 11, line 6 – change request: Replace "Smax14" by plain text.

We changed the sentence: "In the Rhine Basin until Basel, 10-day snowmelt maxima (Smax10) [...]"

**2.3.7 Comment 30**

Page 12, Figure 8 – change request: The horizontal grid lines do only occasionally match the tick marks. Please correct.

We updated Figure 8 and made sure that horizontal grid lines match the tick marks.

**2.3.8 Comment 31**

> Page 13, Figure 9 – change request: The horizontal grid lines do only occasionally match the tick marks. Please correct.

We updated Figure 9 and made sure that horizontal grid lines match the tick marks.

**2.3.9 Comment 32**

> Page 24, Figure B1 – change request: The horizontal grid lines do only occasionally match the tick marks. Please correct.

We updated Figure B1 and made sure that horizontal grid lines match the tick marks.

**2.3.10 Comment 33**

> Page 25, Figure C1 – change request: The horizontal grid lines do only occasionally match the tick marks. Please correct.

We updated Figure C1 and made sure that horizontal grid lines match the tick marks.

**3 Anonymous referee 3**

**3.1 General Comment**

This paper analyses future changes in flood seasonality in the Rhine River Basin at three different global warming levels using the mesoscale Hydrological Model (mHM). The paper is well structured and written, considers earlier work quite well, and provides new insights in flood seasonality changes under climate change for the Rhine basin. Finally, the authors list some next steps to improve the modelling approach as including a glacier module or reservoir and lake functionality.

Thank you very much for reviewing our manuscript. In the following, we provide detailed responses to all your comments. We specifying line numbers, we refer to the marked-up manuscript version below.

**3.2 Specific comments**

**3.2.1 Comment 1**

Data and Methods: suggest to include that the model does not include a glacier and lakes module. For the basin upstream Basel not including lakes can have quite some effect. Now, this becomes only clear at the end of the Conclusions section.

We added this information into the method section: "mHM does not include glacier and lake modules yet." (P6 L35)

**3.2.2 Comment 2**

Page 3, line 15, please describe the downscale and bias correction in more detail. The sentence "adjusts the monthly mean and daily variability of simulated climate data to observations." does not describe how this was done.

We added additional information on the GCM data and the ISI-MIP bias correction approach (P4 L4-13).

**3.2.3 Comment 3**

> Page 3, lines 22-25, The calibration procedure could be described in more detail. 1) For example why was the gauge Lobith also included in the calibration procedure? With MPR, one could have choosen for example three smaller sub-basins to find how well parameters are transferable to the larger basin scale. This makes the calibration more efficient, and would also provide an interesting result (although I understand this is not the focus of the paper, it is an important aspect of this study). 2) What were the specific DDS settings (e.g. number of function evaluations)? Please add these to the text. 3) Finally, how many model parameters were calibrated? At least this gives the reader some insight into the model complexity.

We updated the section on the model calibration and included additional information on the multi-basin approach and specific settings. As MPR enables a very efficient calibration already, we did not see the need to further optimise the calibration routine for our study. Yes, the investigation of the transferability of parameter sets attained in smaller sub-basins to larger basin is an interesting task, however, as you mention, not the focus or our study.

**3.2.4 Comment 4**

> Page 14, line 5. Suggest to change "increased precipitation intensity" to amount, the analysis is about a monthly time scale, so probably better to use amount and not intensity.

We removed the expression 'intensity' and replaced it with either 'sum', 'amount' or talk about antecedent precipitation.

**3.2.5 Comment 5**

> Page 15, line 10 and lines 13-14: Please add this reference as an example: https://agupubs.onlinelibrary.wiley.com/doi/abs/10.1029/2019WR026807, for a mod- elling approach also applied to the Rhine basin that already includes a glacier and lake module.

We included this reference: "Furthermore, the representation of lakes (e.g., Imhoff et al., 2020) [...]".

**3.3 Technical corrections**

**3.3.1 Comment 6**

Fig 7. change "elvation" to "elevation"

We corrected this typo.

**3.3.2 Comment 7**

Page 8, Table 2, change "ration" to "ratio"

We corrected this typo.

**4 Anonymous referee 4**

**4.1 General Comment**

> The paper addresses a highly relevant subject on changes in flood seasonality in the Rhine basin, by analysing how climate change affects different components of the water balance and their aggregated effect on peak flows. The authors elegantly demonstrate by means of model simulations for different GCMs and climate scenarios the contributions of snow melt and rainfall-driven runoff to peak flow generation over the seasons. These analyses form a relevant contribution to earlier studies on the impacts of climate change on peak flows in the Rhine basin, provide nice insight in the underlying contributions of rainfall and snowmelt, and indicate their effects on time shifts in peak flow occurrences. The paper's title well covers the contents; the paper is well-structured, clearly presents its results in text and figures, and the interpretations, discussion and conclusions are well supported by the results. I would rate the significance and quality of the paper 'Good'.

Thank you for reviewing our manuscript. In the following, we provide detailed responses to all your comments.

**4.2 Specific comments**

**4.2.1 Comment 1**

> Fig 1: I would suppose that the nival peak not only shifts to earlier in the season but also becomes smaller under CC - as there will be less total snow accumulation over the winter season.

We thought about this figure the last days. Yes, the point that you mention ("not only shifts to earlier in the season but also becomes smaller") is one of the central aspects we investigate in this study. However, for now, we decided keep this version of the scheme. This helps us formulate our hypothesis in a concise way. As our results indicate, things are more complicated than this simplified scheme. In order to increase clarity with regard to the hypothesis stated at the beginning and the scheme, we included additional information into the discussion (P17 L28–P18 L2).

**4.2.2  Comment 2**

Line 30: 1.5, 2 and 3 degrees warming - relative to 1970 - 2000

We investigate 1.5, 2.0 and 3.0°C global warming levels relative to pre-industrial levels. The period 1971–2000 is assumed to be warmer by 0.46°C compared to pre-industrial levels already. We included additional information into the method section (P7 L3-10).

**4.2.3  Comment 3**

Section 2: provide a bit more information on: which parameters did you calibrate? In particular you have a detailed representation of crops and soil types, did you use reference values in all cases, or did you do any calibration here? How did you choose LAI values for different vegetation types and seasons and latitude? How did you perform the bias correction to GCMs for future climates?

We included more detailed information on the multi-basin calibration approach (P5 L4-13) and extended the description of GCM data and the ISI-MIP bias correction (P4-L3-12). Furthermore, we added more information on the physiographic data sets the form including soils and LAI (P3 L14-22).

**4.2.4  Comment 4**

P4, L11: assess -> assesses

We corrected this typo.

**4.2.5  Comment 5**

P4, L13: bases -> is based

We corrected the sentence.

**4.2.6 Comment 6**

> P6, L6-9. Do I understand here that the projection times of the periods where the 'targeted' warming was reached was different for each realisation? And with different RCPs you may reach the same warming at different moments (e.g. 1.5 degree under RCP 8.0 early in the century, and RCP2.6 only late) - but to what extent are these scenarios different in your simulations (associated P?). Can you indicate which are the according time horizons used in your simulations?

Yes, you understand this correctly. Different GCM/RCP realisations reach the same warming at different time periods. We included additional information on the determination of the 1.5, 2.0 and 3.0 °C time periods. We directly mention the Table S1 in Thober et al. (2018) and included examples of time periods (P7 L3-10). We still hesitate to add the entire table, as it would be a reproduction of entire table already available in Thober et al. (2018).

**4.2.7 Comment 7**

> P6, L14: for the Rhine basin as a whole (e.g. Cologne) a 5 day-period for the precipitation sum seems quite short to generate extreme floods, in particular in view of saturating the soil and travel time of peaks from tributaries.

Yes, due to long travel times, a 5-day window for the entire Rhine basin is (too) short. This was the main reason why we did not conduct this type of analysis for the entire Rhine Basin, but only for sub-basins. In order to investigate runoff components for the entire Rhine Basin, a streamflow component model is necessary (see conclusion P19 L3). As long as there is no streamflow component model available within mHM, we are limited to estimate runoff components on sub-basin level using precipitation and snowmelt directly. Even if this is only a simple estimation of runoff components, we are confident that a 5-day window for precipitation on sub-basin level can provide valuable information.

**4.2.8 Comment 8**

> P6, L19: river discharge at Basel is considerably dampened by the effects of the Swiss lakes. For that reason, earlier studies focused on catchments upstream of the lakes (e.g. Murg, Thur). Can you indicate to what extent timing and maxima of small peaks - in particular after a dry period with low lake levels - are affected by this?

Yes, the large lakes located in Switzerland and Southern Germany affect streamflow characteristics, particularly in the Southern part of the Rhine Basin. We now specifically mention

the lack of a lake module in our model set-up in the method section already (P6 L35). Further-more, we included additional information on the dampening effects in the discussion (P17 L7-10). At this point, we can not provide detailed information on the effect of lakes on characteristics of small streamflow peaks. An analysis of the effect after the implementation of a lake module (run the model with lake module and compare to results from a model set-up without a lake model) seems to be predestined for such a task.

**4.2.9 Comment 9**

> Figure 5: Please indicate on how many runs each histogram is based. From the methods I read how many GCMs reached each warming, with only 8 of them reaching 3 degrees, but this is not clear for the other histograms. To what extent do different numbers of realisation result in different occurrences of highest extremes, and did different GCMs result in different extremes under - in spite of bias correction?

We included the information on the amounts of GCM-RCP realisations reaching each warming level in all figure captions. To get a comprehensive insight, we display results as both boxplots and histograms. We are cautious when interpreting results and focus our discussion exclusively on clear signals, where we can be sure that they are not being caused by different amounts of GCM-RCP realisations. We did not encounter any abnormal differences among GCMs and extremes simulated.

**4.2.10 Comment 10**

> Fig 5; P8, L9: whereas both for Basel and Cochem there is a decline in the timing of summer maxima for higher temperatures, Cologne shows a small peak emerging around DOY 250 - can you explain this? Consider using the same horizontal scale for figs 5d-f.

We now use the same horizontal scale at Fig. 5 d-f. Yes, at gauge Cologne occasionally annual streamflow maxima are recorded in summer around DOY 250. However, taking a close look at the histograms in Fig. 5 f, we conclude that those peaks show up at historic runs and at all warming levels. Yes, there maybe is a slight tendency, but interpreting this as an emerging peak in the distribution goes to far in our opinion. This signal is not clear/robust. Also at gauge Cochem, individual annual streamflow maxima are recorded in summer. This is nothing unusual. We hesitate to draw to much attention on this possible tendency at gauge Colonge and think that it is better to focus the discussion more on the clear signals we attain. Also in regard of the different amount of GCM-RCP realisations (see comment 9) and hence different amount of annual streamflow peaks feeding into the histograms of the different warming levels, it is better to be cautious and not to over-interpret potential signals in the

figures.

**4.2.11 Comment 11**

P10, L4: is detected -> are detected

We corrected this mistake.

**4.2.12 Comment 12**

P10, fig 6 (k,l): by displaying annual maxima distributions we indeed can see how these shift over time, but we cannot see how shifts evapotranspiration maxima link to peak flows, as the connection to the flood events is lost: we cannot see how much was the 'reduction' of the annual peak flow maxima due to evapotranspiration loss (as you do in fig 6 ab indicating the 'contribution' of snowmelt to the annual maxima). It makes sense that under a warming climate annual maximum evapotranspiration goes up - but if that happens in summer when floods never arise it is hard to judge the role of evapotranspiration in changing peak flows. In fig 9g we can see that the contribution of evapotranspiration change is small indeed.

We included the estimated evapotranspiration loss for annual peak flow maxima (Fig. 6 k and l). In addition, we calculate the average annual cycle of ETfrac (Fig. 7 g and h). Yes, it matters whether annual streamflow maxima form during winter or summer. We extended our result and discussion section with the new information on the estimated evapotranspiration loss.

**4.2.13 Comment 13**

P11: Discussions -> discussion

We corrected the section header.

**4.2.14 Comment 14**

P11, L4: ....diminish seasonal snow covers -> please add in a few words the key aspects of that: the total volume, the duration and the timing of melt.

We updated our result and discussions section and provide more detailed information and

numbers on the changing snowmelt characteristics.

**4.2.15 Comment 15**

P11, L6: Smax14 is singular.

We updated Tab. 2 and now define Smax10 as '10-day snowmelt maxima'. We screened through our manuscript and always use this abbreviation in plural.

**4.2.16 Comment 16**

P11, L8: 'forward' -> in first time use, explicitly explain that you mean: 'earlier in the year'

We changed this sentence: "Our results indicate that the detected earlier timing of the annual snowmelt maxima [...]".

**4.2.17 Comment 17**

P11, L10, 13-14: two factors may play a role: the timing of melting, and the amount of snow that has accumulated so far to be available for melting - you do not indicate the maximum amount of snow that has accumulated by the end of the season to be available for melt.

We updated our discussion section and included more specific information on changes in snowmelt characteristics (e.g., P17 L31). .

**4.2.18 Comment 18**

P11, L15. I do not see a contradiction suggested by using 'however'. Actually, you change subject here to low flow situations, as caused by disappearing glaciers and intensified evapotranspiration, which becomes different from 'lower maxima',

In this case, we do not try to suggest a contradiction. We use 'however' in the sense of 'in spite of that/on the other hand'. We try to say that changes in snow cover do not increase flood risk, however, they might aggravate low-flows. Yes, we use the 'however' here to connect/switch

focus from one subject (influence changes snowmelt on flood hazard) to the subject 'potential influences of changes in snow cover on low-flow'. In our opinion, it is important to shortly mention here that the changes in snow cover affect both high and low flow.

**4.2.19 Comment 19**

> P12. l6 (and in the rest of the paper, in particular in the conclusion, P14, L32): I am not sure whether you should formulate this as 'intense rainfall events' in mm per hour and use this formulation for both summer and winter. 'High intensity' rather relates to high intensity summer storms - as you indicate in lines 12-15 here, but for the winter season I would not formulate that as intensity. Moreover, you consider in your study accumulated precipitation sums over 5 days - that is an amount, not an intensity.

To avoid any misunderstandings, we removed the expression "intensity" and replaced it with either 'sum', 'amount' or talk about antecedent precipitation.

**4.2.20 Comment 20**

> P12, L12. Here you make a relevant statement - that the summer extremes were still supported by snow melt from the Alps to produce their maxima - I presume that that has been derived from the associated historic descriptions. With reduced snow melt in late spring this would indeed reduce the risk of summer floods. Conversely, higher temperatures under future climate change may lead to more intense summer precipitation - still causing higher peak flows. Here we may encounter the questions: does the latter only hold for smaller catchments within the basin, or do these still feed large floods to Cologne? And some GCMs show a N-S difference in precipitation change (some even the signal) across the Rhine basin: is that the case in your experiments? From the histograms in fig 6g this is hard to derive. It would affect probabilities of extreme summer floods, as we experienced in early 2000s in the Elbe.

Yes, this is a very important point. For large basins, such as gauge at Cologne, local convective storms hardly play any role. Our results indicate an increase in 5-day rainfall amounts in the investigated sub-basins, also for summer. We updated our manuscript and write more detailed on 'counterbalancing effects' between changes in snowmelt and precipitation.

**4.2.21 Comment 21**

> P14, L3: It is not a true 'interaction' between snow fall and precipitation, but their effects are counterbalancing - as you explain in the following lines.

We removed the expression 'interaction' in this regard and now describe it as write about a counterbalancing effect'.

**4.2.22 Comment 22**

> P14, L9: hint -> suggest

We changed the sentence: "Simulating the Rhine River for the time frame 1901-2006, Stahl et al. (2016) suggest [...]" (P17 L22).

**4.2.23 Comment 23**

> P14, L14: originate - > originates; During this period, we have experienced already over 0.5 degree of warming, so it would be interesting to know what the average for the past few decades would be.

We corrected the typo. We agree, the numbers we present from Stahl et al. (2016) refer to the historic period 1901–2006. Stahl et al. (2006) provide both long-term averages and figures with the time series of the runoff components: Fig. 4 in
https://chr-khr.org/en/file/1057/download?token=Zg6SY04i
Yes, due to rising temperatures within the 20th century, there possibly has been changes in the fraction of snowmelt contributions to runoff already. Stahl et al. (2016) suggest that "it is difficult to detect uniform long-term changes governing the entire Rhine basin". In our opinion, Fig. 4 also highlights the strong annual to decadal variability of the relative fraction of the streamflow components. We included this information into the revised manuscript (P17 L24-25).

**4.2.24 Comment 24**

> P14, L33: 'intense precipitation events' (see earlier comment): would you describe the precipitation driving the Elbe floods earlier this millennium as 'high-intensity' or 'large amounts'? (actually, I think it was a combination of both....). I would avoid suggesting that more intense summer showers will cause the Rhine to flood Cologne.

Yes, to get large "amounts" a high "intensity" usually is necessary. To avoid any misunderstandings, we removed the expression 'intensity' and replaced it with either 'sum', 'amount' or talk about antecedent precipitation.

**4.2.25 Comment 25**

> Discussion point to consider: Your 30-year time slices from which you determined the flood maxima may not include the very extremes that are relevant for flood protection - in the box plot you see a few isolated extremes that occurred in your simulations. To what extent do you think this affects your overall message?

Yes, with regard to flood protection, the very high values are of great interest. In this analysis, we focused on the hypothesis of merging the flood regimes that might result in the creation of such extreme events. We are mostly interested in temporal shift and changes in magnitudes. In this regard, we focus on changes in underlying flood-generating processes. "Isolated extremes" do not influence our results and overall message. We included information on the still pending analysis of isolated peaks in the discussion: "A detailed analysis of isolated extreme simulated is still pending." (P18 L2)

**4.2.26 Comment 26**

> P15, L10. Under recommendations: Would calibration on observed extent of snow cover using RS support calibration of snow melt modules to support these analyses? To what extent would precipitation falling on a snow cover further enhance melting of a snow cover (so: not snow melt not only depends on T, but also on warm precipitation water) - and would we need to consider that in modeling? Lakes: of course these buffer flow in dry periods, but that was not the focus of your paper, it might be interesting to see their role in generating peak flows.

Yes, the usage of satellite-based snow cover maps during model calibration and/or validation can further improve the simulation of the snow cover. We included this information

in the the conclusion (P19 L1-2). The snow module currently available in mHM is based on a degree-day approach. As we describe in the method section: "In order to account for snowmelt following the energy input from liquid rainfall, degree-day factors are increased depending on the amount of liquid precipitation. Degree-day factors only can increase to a certain threshold value." Hence, mHM already has a (simple) way of addressing the energy input through liquid rain. The implementation of a physically-based snow routine might improve this aspect and addresses rain-on-snow events better. The implementation of a snow physically based snow routine represents one possible next step to improve the hydrological simulations (P18 L32).

**4.2.27 Comment 27**

> For policy makers / river managers it may be relevant to see a conclusion on whether we should anticipate changes in summer floods - as the use of the river banks (agriculture, tourism) has a strong seasonality.

We updated our discussion and now we also mention changes in summer.

**5 Marked-up manuscript version**

Marked-up manuscript version produced using "latexdiff" on the following pages. It compares HESSD discussion manuscript and the revised version.

[revised manuscript text omitted]

Zemp, M., Haeberli, W., Hoelzle, M., and Paul, F.: Alpine glaciers to disappear within decades?, Geophysical Research Letters, 33, https://doi.org/10.1029/2006GL026319, 2006.

[Figure]

**Figure A1.** Scatter plot of observed and simulated annual streamflow maxima (MAX) and the 90 % streamflow quantile (Q90) of the hydrological year starting 1 October for all validation gauges (a-d; Fig. 2) and for selected gauges (e-h). Panels a, b, e and f depict observed discharge and simulated discharge using E-OBS-based meteorological forcing. Panels c, d, g and h depict observed discharge and simulated discharge using climate model data from the ISI-MIP project. Time frame investigated: 1951–2000.

[Figure]

**Figure B1.** Timing of annual streamflow maxima observed and simulated using E-OBS-based meteorological forcing and climate model data from the ISI-MIP project for all validation gauges (Fig. 2). Time frame investigated: 1951–2000.

[Figure]

**Figure C1.** Streamflow quantiles (90 %) for every month of the year based on daily resolution observations and simulations using E-OBS-based meteorological forcing and climate model data from the ISI-MIP project for all validation gauges (Fig. 2). Time frame investigated: 1951–2000.

---

## Author Response (AR2)

**hess-2020-605**
**Responses to the editor**

**Erwin Rottler, Axel Bronstert, Gerd Bürger and Oldrich Rakovec**

March 23, 2021

Dear Mr. Weiler,

thank you very much for taking a look at the revised version of our manuscript. We are very grateful for your suggestions. In the following, we provide detailed responses to your comments.

On behalf of all authors,

Sincerely,

Erwin Rottler

**Contents**

**1 Comment 1**

> The definition of Sfrac (snowmled contributions) seem highly arbitrary to me (why 10 days for snowmelt and only 5 days for rainfall). How are the different phases (rain vs snow) of precipitation considered. Looking for example at the simulation and discussions done in https://doi.org/10.1002/hyp.11361, I think there should be either a clear demonstration that this ratio is a good representation of the snowmelt contribution or these results should better be removed from the paper, as otherwise these results are considered to be a correct representation of snowmelt contribution in the Rhine. At the moment, the simulations of Sfrac are not even compared to other approaches done for the Rhine valley.

Yes, we agree that we need to better justify and discuss the calculation of $S_{frac}$. It is not a direct quantification of the different streamflow components, but only a simple ratio calculated based on snowmelt and liquid precipitation and needs to be presented as such. As a first step, we changed the definition presented in Table 2. We do not refer to $S_{frac}$ as an estimate of streamflow components anymore, but only as a ratio that provides first indications of changes in the importance of snowmelt compared to liquid precipitation. We changed corresponding sentences in the manuscript (e.g., Page 8 Line 10, Page 9 Line 13 and 17, Page 11 Line 8, Captions Figure 6 and 7) and include information on the simple estimate we use and the potential of streamflow component models in the discussion (Page 16 Line 25-28). In the definition as well as other parts of the manuscript we made sure to always state that for the calculation of this ration we use **liquid** precipitation only. At Page 16 Line 21-25, we provide numbers attained by Stahl et al. (2016). In the first paragraph of section 2.2 (Page 7 Line 8 - Page 8 Line 4) we explain the selection of 5/10 day windows. The selection bases on information from previous studies and personal experience. For consistency reason we base the calculation of $S_{frac}$ and $ET_{loss}$ on values with same window sizes.

**2 Comment 2**

> I think that the model is currently limited in its results as glaciers are not considered at all. This is mentioned in the manuscript, but I think this should already be mentioned in the abstract, as this is a major limitation of the model in its current set-up.

We included this information into the abstract: " To refine attained results, next steps need to be the representation of glaciers [...]" (Page 1 Line 15).

**3 Comment 3**

> The missing independent validation of the snow routine using satellite based snow cover maps is now mentioned, but only in the conclusion. The potential should also be discussed as well as already mentioned in the abstract.

We included the potential of satellite-based snow cover maps to calibrate/validate model simulations into the abstract and the discussion: "To refine attained results, next steps need to be [...] an independent validation of the snow routine using satellite based snow cover maps." (Page 1 Line 15-17) and "Next step to improve the representation of snow accumulation and melt in the model set-up needs to the usage of satellite-based snow cover maps in a multi-criteria-calibration and/or for an independent validation of the snow routine" (Page 14, Line 8-10).

**4 Marked-up manuscript version**

Marked-up manuscript version produced using "latexdiff" on the following pages. It compares the re-submitted manuscript after minor revision and the revised version following additional remarks of the editor.

**Projected changes in Rhine River flood seasonality under global warming**

Erwin Rottler[1], Axel Bronstert[1], Gerd Bürger[1], and Oldrich Rakovec[2,3]

[1]Institute of Environmental Science and Geography, University of Potsdam, Karl-Liebknecht-Straße 24–25, 14476 Potsdam, Germany
[2]UFZ-Helmholtz Centre for Environmental Research, Permoserstraße 15, 04318 Leipzig, Germany
[3]Faculty of Environmental Sciences, Czech University of Life Sciences Prague, Kamýcká 129, Praha – Suchdol, 165 00, Czech Republic

**Correspondence:** Erwin Rottler (rottler@uni-potsdam.de)

**Abstract.** Climatic change alters the frequency and intensity of natural hazards. In order to assess potential future changes in flood seasonality in the Rhine River Basin, we analyse changes in streamflow, snowmelt, precipitation, and evapotranspiration at 1.5, 2.0 and 3.0 °C global warming levels. The mesoscale Hydrological Model (mHM) forced with an ensemble of climate projection scenarios (five general circulation models under three representative concentration pathways) is used to simulate the present and future climate conditions of both, pluvial and nival hydrological regimes.

Our results indicate that future changes in flood characteristics in the Rhine River Basin are controlled by increases in antecedent precipitation and diminishing snow packs. In the pluvial-type sub-basin of the Moselle River, an increasing flood potential due to increased antecedent precipitation encounters declining snowpacks during winter. The decrease in snowmelt seems to counterbalance increasing precipitation resulting in only small and transient changes in streamflow maxima. For the Rhine Basin at Basel, rising temperatures evoke changes from solid to liquid precipitation, which enhance the overall increase in precipitation sums, particularly in the cold season. At gauge Basel, the strongest increases in streamflow maxima show up during winter, when strong increases in liquid precipitation encounter almost unchanged snowmelt-driven runoff. The analysis of snowmelt events for gauge Basel suggests that at no point in time during the snowmelt season, a warming climate results in an increase in the risk of snowmelt-driven flooding. Snow packs are increasingly depleted with the course of the snowmelt season. We do not find indications of a transient merging of pluvial and nival floods due to climate warming. To refine attained results, next steps need to be the representation of glaciers and lakes in the model set-up, the coupling of simulations to a streamflow component model and an independent validation of the snow routine using satellite-based snow cover maps.

[revised manuscript text omitted]

10 for an independent validation of the snow routine.

    Our results confirm previous studies suggesting that rising temperatures might lead to stronger precipitation events (e.g., Lehmann et al., 2015; Alfieri et al., 2015; King and Karoly, 2017; Bürger et al., 2019; Rottler et al., 2020) (Fig. 6 g-j and Fig. 9 c-f) and a shift from solid to liquid rainfall (e.g., Allamano et al., 2009; Addor et al., 2014; Davenport et al., 2020) (Fig. 7 a and b). In catchments having mixed hydrological regimes with rainfall and snowmelt, rising temperatures seem to lead to a

[Figure]

**Figure 9.** Magnitudes of 10-day snowmelt maxima ($S_{max10}$; a and b), liquid (c and d) and total (e and f) 5-day precipitation ($P_{max5}$) and 10-day actual evapotranspiration maxima ($ET_{max10}$; g and h) for sub-basins at Basel and Cochem under selected warming levels (14 GCM-RCP realisations reach 1.5 °C, 13 reach 2 °C and 8 reach 3 °C warming). Whiskers and outliers of the boxplots are not displayed.

shift from snowmelt to rainfall as most important flood generating process (Vormoor et al., 2015, 2016). Reconstructing the largest floods in the High Rhine since 1268, Wetter et al. (2011) indicate that about half of all large events occurred during summer due heavy precipitation combined with high baseflow from snow- and ice-melt. Our results indicate that with rising temperatures, most flood events will occur in winter (Fig. 5 d).

In March and April, the increase in rainfall amounts in the basin at Basel compares to increases in winter, the magnitudes of streamflow maxima, however, hardly change (Fig. 8 a). We suggest that the increasing potential of rainfall-induced flooding is counterbalanced by decreasing snowmelt (Fig. 9 a and c). Furthermore, our results hint at a transient increase in flood magnitudes during May and June (Fig. 8 a). It seems that during those two months, snowmelt is still strong enough to support an increase in streamflow peaks due to increased antecedent precipitation at moderate warming levels (1.5 °C and 2.0 °C). With further rising temperatures, however, the magnitudes of streamflow maxima reduce along with declining snowmelt (Fig. 8 a). The mHM model set-up that we use to simulate the Rhine River does not include a lake module. The simulation results attained for the Rhine Basin, particularly for gauge Basel, can be further refined by the representation of the large lakes located in Switzerland and Southern Germany (Imhoff et al., 2020). The large storage volume and the possibility to regulate lake levels dampen streamflow peaks.

For gauge Cochem and the associated sub-basin of the Moselle River, we detect similar counterbalancing effects between snowmelt and rainfall: an increasing flood potential due to increased precipitation amounts encounters declining snow packs. Again, decreases in snowmelt magnitudes seem to counterbalance increased precipitation resulting in comparatively small and transient increases in streamflow maxima (Fig. 8 b and Fig. 9 b and d). As highest mountains in the sub-basin only reach up to around 1300 m a.s.l., snowmelt compensation effects, i.e., snowmelt from higher elevations, at least partly, replaces the lack of snowmelt from lower elevation, only plays a marginal role. Analysing changes in frequencies of rain-on-snow (RoS) events with flood-generating potential for large parts of Europe for the historic time frame 1950–2011, Freudiger et al. (2014) hint at similar processes changing flood hazard. Their analyses suggest an increase in flood hazard from RoS events in medium-elevation mountain ranges in the Rhine River Basin in winter due to increased rainfall and a decrease in RoS events in spring due to decreases in snow cover. Although important Rhine tributaries, such as the Moselle River, often are characterised as pluvial-type rivers, the importance of snowmelt as runoff component must not be underestimated. Simulating the Rhine River for the time frame 1901–2006, Stahl et al. (2016) suggest that at gauge Cochem, 26 % of the annual streamflow originates from snowmelt. During winter, this fraction increases up to almost 40 % (see also Fig. 7 b). However, the inter-annual variability of annual streamflow and the relative fractions of streamflow components is high, particularly in pluvial-type tributaries of the Rhine River (Stahl et al., 2016). The simple estimate $S_{frac}$, which is based on the amount of snowmelt and liquid precipitation, provides the first-order approximation of future changes in the importance of snowmelt during peak flow formation and as streamflow component. However, to correctly quantify changes in streamflow components, the coupling of simulations to a streamflow component model is required (Stahl et al., 2016; Weiler 
[revised manuscript text omitted]

Weiler, M., Seibert, J., and Stahl, K.: Magic components—why quantifying rain, snowmelt, and icemelt in river discharge is not easy,
Hydrological Processes, 32, 160–166, https://doi.org/https://doi.org/10.1002/hyp.11361, 2018.

Wetter, O., Pfister, C., Weingartner, R., Luterbacher, J., Reist, T., and Trösch, J.: The Largest Floods in the High Rhine Basin since 1268 Assessed from Documentary and Instrumental Evidence, Hydrological Sciences Journal, 56, 733–758, https://doi.org/10.1080/02626667.2011.583613, 2011.

Zemp, M., Haeberli, W., Hoelzle, M., and Paul, F.: Alpine glaciers to disappear within decades?, Geophysical Research Letters, 33, https://doi.org/10.1029/2006GL026319, 2006.

[Figure]

**Figure A1.** Scatter plot of observed and simulated annual streamflow maxima (MAX) and the 90 % streamflow quantile (Q90) of the hydrological year starting 1 October for all validation gauges (a-d; Fig. 2) and for selected gauges (e-h). Panels a, b, e and f depict observed discharge and simulated discharge using E-OBS-based meteorological forcing. Panels c, d, g and h depict observed discharge and simulated discharge using climate model data from the ISI-MIP project. Time frame investigated: 1951–2000.

[Figure]

**Figure B1.** Timing of annual streamflow maxima observed and simulated using E-OBS-based meteorological forcing and climate model data from the ISI-MIP project for all validation gauges (Fig. 2). Time frame investigated: 1951–2000.

[Figure]

**Figure C1.** Streamflow quantiles (90 %) for every month of the year based on daily resolution observations and simulations using E-OBS-based meteorological forcing and climate model data from the ISI-MIP project for all validation gauges (Fig. 2). Time frame investigated: 1951–2000.